# Graph-Preference Learning: Debiasing Network-Sampled Human Feedback for Target Welfare Estimation

Guangrui Fan [1 2]   Dandan Liu [3]   AZNUL QALID MD SABRI [2]   Lihu Pan [1]

## Abstract

Preference-based reward modeling is a core component of RLHF and DPO pipelines. In practice, the humans providing preference feedback are rarely an i.i.d. sample: recruitment and exposure often follow social, institutional, or spatial structure, inducing non-uniform inclusion probabilities that correlate with graph centrality. We formalize preference learning with *network-sampled* annotators and show that identity-agnostic scalar reward modeling implicitly represents an inclusion-weighted welfare, over-representing structurally central communities when the inclusion distribution $q$ differs from a designer-chosen target weighting $\pi$. We propose Graph-Preference Learning, which combines (i) a graph-personalized reward model that shares statistical strength across neighboring annotators and (ii) graph-balanced aggregation that computes stabilized importance weights to target $\pi$. Our analysis characterizes the induced welfare represented by the learned aggregate reward and bounds its deviation from the target in terms of weight mismatch, reward-model approximation, and finite-sample effects. Experiments on synthetic graphs and a *semi-synthetic* case study on the LMArena preference dataset, where biased inclusion is *induced* via graph-based sampling, demonstrate up to 62% reduction in target-welfare recovery error and 17% reduction in cross-language performance gaps under biased inclusion[1].

## 1. Introduction

Reinforcement learning from human feedback (RLHF) and direct preference optimization (DPO) are widely used to adapt models to human preferences and norms. A key component shared by these pipelines is *preference-based reward modeling*: learning a scalar score from pairwise comparisons and using it to rank candidate outputs and/or guide downstream optimization (Ouyang et al., 2022; Rafailov et al., 2023; Kaufmann et al., 2025; Lambert, 2025). In this work we focus on this reward/preference-model stage: *what aggregate reward signal is learned when preference feedback is collected under non-uniform inclusion, and how can we debias it toward a designer-chosen welfare target?*

In many real deployments, feedback providers are not an i.i.d. sample from a homogeneous pool. Recruitment and exposure are mediated by platforms and institutions, and can propagate through social, institutional, or spatial structure. Such mechanisms induce *inclusion probabilities* that correlate with graph centrality: structurally central communities contribute disproportionately many comparisons. When annotator identity is discarded (as in standard scalar reward modeling), the learned reward implicitly represents an inclusion-weighted welfare, potentially over-representing central groups. This phenomenon is closely related to classical results in network sampling and graph fairness (Wejnert, 2010; Lerman et al., 2016; Saxena et al., 2024; Dong et al., 2022).

We address this gap by treating feedback providers as nodes in a graph and explicitly distinguishing the *observed* inclusion distribution $q$ from a *target* welfare weighting $\pi$. We propose Graph-Preference Learning, a framework combining graph-personalized reward modeling with graph-balanced aggregation to learn an aggregated score $\bar{r}_\theta(x, y)$ whose induced welfare approximates the target.

In this paper, we formalize reward modeling with *network-sampled* annotators by distinguishing inclusion $q$ from welfare target $\pi$ on an annotator graph. We show that naive scalar reward modeling minimizes a $q$-mixture risk (Theorem 4.1) and propose Graph-Preference Learning—graph-personalized rewards (GPRM) plus stabilized aggregation (GBA)—to target $\pi$. We bound welfare deviation in terms

[1]School of Computer Science, Taiyuan University of Science and Technology, Taiyuan, Shanxi, China [2]Faculty of Computer Science and Information Technology, Universiti Malaya, KL, Malaysia [3]Department of Media and Communication Studies, Universiti Malaya, KL, Malaysia. Correspondence to: Guangrui Fan <fgr@tyust.edu.cn>.

*Proceedings of the 43rd International Conference on Machine Learning*, Seoul, South Korea. PMLR 306, 2026. Copyright 2026 by the author(s).

[1]We discuss how the resulting weights and aggregate reward can be integrated into downstream DPO/PPO pipelines in the appendix.

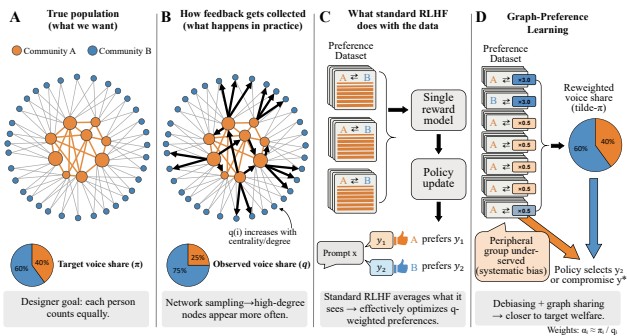

*Figure 1.* **Network-induced inclusion bias in preference-based reward modeling.** (A) Annotators form a graph with communities; the designer specifies a normative weighting $\pi$ (e.g., equal-per-person or equal-per-community). (B) Feedback collected via networked mechanisms (platform exposure, random-walk/snowball recruitment) induces a non-uniform inclusion distribution $q$ that can over-sample central nodes/communities. (C) Standard identity-agnostic reward modeling trains on the marginal data, implicitly representing a $q$-weighted aggregate preference. (D) Graph-Preference Learning (i) learns personalized rewards $r_\theta(x, y, i)$ with graph-based sharing and (ii) computes stabilized importance weights $\alpha_i^\star$ to approximate a target welfare, yielding induced weights $\tilde{\pi}$ closer to $\pi$ and an aggregated score $\bar{r}_\theta(x, y)$.

of $\|\tilde{\pi} - \pi\|_1$, model approximation, and finite-sample effects. On synthetic graphs and a *semi-synthetic* LMArena case study with induced network-sampled inclusion, we achieve up to 62% welfare-error reduction, 18% accuracy improvement, and 17% cross-language gap reduction. All evaluations are at the reward-model level; full related work is in Appendix A.

**Conflict of Interest Disclosure.** All authors are affiliated solely with academic institutions (Taiyuan University of Science and Technology and Universiti Malaya), and none of the authors hold employment, consulting positions, equity, or other financial relationships with companies whose products or models are evaluated in this work. The semi-synthetic case study is conducted on the publicly released LMArena preference dataset (Chiang et al., 2024), which we use only as a source of human preference annotations; we do not evaluate, endorse, or compare any specific commercial model developed by an entity associated with the authors. No external entity influenced the design, analysis, or reporting of the experiments. The authors have no financial conflicts of interest relevant to the contributions of this paper.

## 2. Problem Formulation

This section formalizes the setting of preference-based reward modeling with networked annotators. We introduce the annotator graph, define inclusion and welfare distributions, and characterize the gap between observed and target welfare.

### 2.1. Preference Feedback and Decision Rules

Let $\mathcal{X}$ denote the space of inputs (e.g., prompts) and let $\mathcal{Y}_x$ denote the set of candidate outputs available for input $x \in \mathcal{X}$. A *decision rule* $\mu_\phi$ is any conditional distribution $\mu_\phi(y \mid x)$ over $y \in \mathcal{Y}_x$. In this paper we learn a scalar reward/preference score used for preference prediction and offline reranking.

Human feedback is collected as pairwise preferences over candidate outputs. Concretely, we observe a dataset

$$\mathcal{D} = \left\{ (x_t, y_t^+, y_t^-, i_t) \right\}_{t=1}^{T}, \tag{1}$$

where $x_t \in \mathcal{X}$ is an input, $y_t^+, y_t^- \in \mathcal{Y}_{x_t}$ are two candidate outputs, and $i_t \in V$ is the identity of the annotator who expressed $y_t^+ \succ y_t^-$.[2] We keep annotator identities $i_t \in V$ explicit (often discarded by standard scalar reward modeling).

### 2.2. Networked Annotators and Inclusion

We model annotators as nodes in a graph capturing social, institutional, or spatial relationships.

**Definition 2.1** (Annotator graph). Let $G = (V, E)$ be an undirected graph, where $V = \mathcal{I}$ is the set of annotators and $E \subseteq V \times V$ encodes relationships (e.g., social ties, shared institution, spatial adjacency, or similarity). We write $d_i = \deg(i)$ for node degree and $N(i)$ for neighbors of $i$.

In practice, $G$ may be constructed from explicit ties or similarity between annotators.

**Network-induced inclusion.** Feedback collection mechanisms are frequently *non-uniform* over $V$: platform exposure over-samples active users, recruitment can propagate through social ties, and institutions contribute proportionally to size or visibility. We model this using an *inclusion distribution* $q$ over annotators.

**Definition 2.2** (Inclusion probabilities). At each feedback event $t$, the annotator index $i_t$ is sampled from a fixed distribution $q(\cdot)$ on $V$, called the *inclusion distribution*. We denote $q_i = q(i)$ so that $\sum_{i \in V} q_i = 1$, and write

$$i_t \sim q(\cdot), \qquad (x_t, y_t^+, y_t^-) \sim \mathcal{T}(i_t), \tag{2}$$

where $\mathcal{T}(i)$ is a (task-dependent) distribution over prompts and candidate outputs conditional on annotator $i$.

In many network sampling mechanisms (e.g., random-walk or respondent-driven sampling), $q$ correlates with graph centrality (often approximately $q_i \propto d_i$). We do not require knowing the exact mechanism and allow $q$ to be unknown but estimable from logged participation.

---

[2]We restrict to pairwise comparisons for clarity; extensions to graded feedback or top-$k$ rankings are straightforward.

### 2.3. Individual Utilities and Welfare Targets

We assume each annotator $i$ has a latent utility function over outputs.

**Definition 2.3** (Individual utilities). For each $i \in V$, let $u_i : \mathcal{X} \times \mathcal{Y}_x \to \mathbb{R}$ denote annotator $i$'s latent utility, where $u_i(x, y)$ measures the (unobserved) desirability of output $y$ for input $x$ according to annotator $i$. We assume observed pairwise preferences are consistent with these utilities; see Appendix B.4 for a logistic choice model example.

**Target welfare weights.** A central modeling choice is a *normative* weighting $\pi$ over annotators that specifies whose utilities should count and how much.[3]

**Definition 2.4** (Target social welfare). We consider linear welfare functionals of the form

$$W_{\text{target}}(\mu_\phi) = \sum_{i \in V} \pi_i U_i(\mu_\phi), \qquad \pi_i \geq 0, \qquad \sum_{i \in V} \pi_i = 1,$$
(3)

where $U_i(\mu_\phi) = \mathbb{E}_{x \sim p_{\text{data}}, y \sim \mu_\phi(\cdot|x)}[u_i(x, y)]$ is annotator $i$'s expected utility under decision rule $\mu_\phi$.

Equivalently, at the *per-output* level we can define the $\pi$-weighted welfare score

$$w_\pi(x, y) := \sum_{i \in V} \pi_i u_i(x, y),$$
(4)

so that $W_{\text{target}}(\mu_\phi) = \mathbb{E}_{x, y \sim \mu_\phi}[w_\pi(x, y)]$. Canonical choices include equal-per-person ($\pi_i = 1/|V|$) and equal-per-community ($\pi_i = 1/(K|C_{k(i)}|)$).

**Observed sampling vs. normative target.** We distinguish the *observed* inclusion distribution $q$ induced by the collection pipeline from a *normative* target weighting $\pi$ chosen by a designer or governance process. Typical targets include equal-per-person and equal-per-community/region; a boxed summary appears in Appendix B.1.

### 2.4. What Standard Preference-Based Reward Modeling Learns

Standard (identity-agnostic) reward modeling discards annotator identity and fits a scalar score $r_\theta(x, y)$ to the marginal preference distribution induced by sampling $i \sim q$. As shown in Theorem 4.1, this corresponds to minimizing a $q$-mixture objective; under standard welfare-consistency conditions, the learned score admits the interpretation of an *observed* (inclusion-weighted) welfare score

$$w_q(x, y) := \sum_{i \in V} q_i u_i(x, y),$$
(5)

and, at the decision-rule level,

$$W_{\text{obs}}(\mu_\phi) := \sum_{i \in V} q_i U_i(\mu_\phi).$$
(6)

Assumption B.1 and a mixture-of-logits approximation bound appear in Appendix B.4 and Appendix D.3.

### 2.5. Network-Induced Welfare Gap

When $q \neq \pi$, the welfare implicit in the observed data differs from the designer-chosen target welfare. This mismatch can be expressed both at the per-output level and at the decision-rule level.

**Definition 2.5** (Network-induced welfare gap). For any $(x, y)$, define the welfare gap

$$\Delta(x, y) := w_\pi(x, y) - w_q(x, y) = \sum_{i \in V}(\pi_i - q_i) u_i(x, y).$$
(7)

For any decision rule $\mu_\phi$, the induced gap is

$$\Delta(\mu_\phi) := W_{\text{target}}(\mu_\phi) - W_{\text{obs}}(\mu_\phi) = \sum_{i \in V}(\pi_i - q_i) U_i(\mu_\phi).$$
(8)

When $q$ correlates with graph centrality (e.g., $q_i \propto d_i$), structurally central annotators and communities can be over-represented relative to the target $\pi$, yielding systematic welfare distortions.

### 2.6. Goal: Target-Welfare Reward Modeling

Our goal is to learn a reward/preference model that represents the *target* welfare rather than the *observed* inclusion-weighted welfare. Concretely, we seek to construct (i) a personalized reward model $r_\theta(x, y, i) \approx u_i(x, y)$ and (ii) an aggregated reward $\bar{r}_\theta(x, y)$ whose induced welfare weights $\tilde{\pi}$ satisfy $\tilde{\pi} \approx \pi$. This enables *target-welfare* candidate ranking and welfare estimation directly at the reward-model level. Downstream use in RLHF/DPO (e.g., reweighting preference optimization objectives or substituting $\bar{r}_\theta$ as the reward) is a natural application but is orthogonal to our main empirical evaluation and is discussed in the appendix.

## 3. Method

Graph-Preference Learning comprises two components: **(1) Graph-personalized reward modeling (GPRM)** learns $r_\theta(x, y, i) \approx u_i(x, y)$ while sharing statistical strength across neighbors in $G$; **(2) Graph-balanced aggregation (GBA)** computes stabilized importance weights $\alpha_i^\star$ from $(\pi, \hat{q}, G)$ to define induced welfare $\tilde{\pi}_i := \hat{q}_i \alpha_i^\star$ and aggregated reward $\bar{r}_\theta(x, y) = \sum_i \tilde{\pi}_i r_\theta(x, y, i)$. Given an annotator graph $G = (V, E)$, preference data $\mathcal{D}$ with annotator IDs, target weights $\pi$, and inclusion estimator $\hat{q}$, the method

returns a personalized reward $r_\theta(x, y, i)$ and an aggregated score $\bar{r}_\theta(x, y)$ with $\tilde{\pi} \approx \pi$.

### 3.1. Graph-Personalized Reward Model (GPRM)

GPRM learns a per-annotator reward (utility proxy) while leveraging the graph $G$ to encourage local smoothness and improve sample efficiency when each annotator contributes limited feedback.

Each annotator $i \in V$ is associated with an initial feature vector $h_i \in \mathbb{R}^{d_0}$ (e.g., demographic attributes, interaction statistics, or a learned ID embedding). We compute graph-aware annotator embeddings via a message-passing encoder $g_\psi$:

$$v_i = g_\psi(i; G, \{h_j\}_{j \in V}), \qquad i \in V, \qquad (9)$$

where $g_\psi$ may be instantiated as a GCN/GraphSAGE-style network. Let $\mathbf{V} \in \mathbb{R}^{|V| \times d}$ denote the matrix of embeddings.

For each input–output pair $(x, y)$ we compute a representation $\phi(x, y) \in \mathbb{R}^{d_x}$ (e.g., from a frozen backbone encoder plus a small MLP). The personalized reward model is

$$r_\theta(x, y, i) = f_\theta(\phi(x, y), v_i), \qquad (10)$$

where $f_\theta$ maps $[\phi(x, y); v_i]$ to a scalar. We optionally incorporate an annotator-specific scale $s_i$ to model rater strictness; see Appendix C.1.

Given preference tuples $(x_t, y_t^+, y_t^-, i_t)$, we fit $r_\theta$ using a Bradley–Terry (logistic) preference likelihood with *importance weights* $\alpha_{i_t}^\star$ supplied by GBA (Section 3.2):

$$\mathcal{L}_{\text{GPRM}}(\theta, \psi) = -\sum_{t=1}^{T} \alpha_{i_t}^\star \log \sigma\Big(r_\theta(x_t, y_t^+, i_t) -$$

$$r_\theta(x_t, y_t^-, i_t)\Big) + \lambda_v \operatorname{Tr}(\mathbf{V}^\top L_w \mathbf{V}) + \lambda_\theta \|\theta\|_2^2, \quad (11)$$

where $L_w$ is a (weighted) graph Laplacian and $\lambda_v, \lambda_\theta$ are hyperparameters (an optional rater-scale extension is in Appendix C.1). When $\alpha_i \approx \pi_i/q_i$, the weighted objective behaves as if annotators were sampled from the target welfare distribution $\pi$ rather than the observed inclusion $q$. In strongly heterophilous graphs, large $\lambda_v$ can over-smooth genuine preference differences; practical mitigations are discussed in Appendix C.1.

### 3.2. Graph-Balanced Aggregation (GBA)

GBA computes stabilized importance weights $\alpha_i^\star$ that (i) debias learning toward the target welfare $\pi$ and (ii) define an induced welfare distribution $\tilde{\pi}$ used to aggregate personalized rewards into a single scalar score.

We estimate $\hat{q}$ from participation counts with smoothing; alternative estimators (degree-/centrality-based or hybrid) are described in Appendix C.2.

A direct debiasing rule is the inverse propensity ratio

$$\tilde{\alpha}_i = \frac{\pi_i}{\hat{q}_i + \varepsilon}, \qquad (12)$$

with $\varepsilon > 0$ for stability. However, raw ratios can have high variance and can be sensitive to noise in $\hat{q}$.

We compute stabilized weights $\alpha$ by solving the following convex QP:

$$\alpha^\star = \arg \min_{\alpha \in \mathbb{R}^{|V|}} \left\{ \frac{1}{2} \|\alpha - \tilde{\alpha}\|_2^2 + \lambda_{\text{smooth}} \, \alpha^\top L_w \alpha \right.$$

$$\left. + \lambda_{\text{comm}} \sum_{k=1}^{K} \Big( \sum_{i \in C_k} \hat{q}_i \alpha_i - \rho_k \Big)^2 + \lambda_{\text{var}} \sum_{i \in V} \hat{q}_i \alpha_i^2 \right\} \text{s.t.}$$

$$\alpha_i \geq 0 \, \forall i \in V, \quad \sum_{i \in V} \hat{q}_i \alpha_i = 1,$$

$$(13)$$

The objective trades off closeness to the raw ratios, graph smoothness via $L_w$, desired community masses $\rho_k$, and weight-variance (ESS) control. We solve Eq. (13) once as preprocessing; solver details are in Appendix C.3 and pseudocode appears in Algorithm 3.

Given stabilized weights $\alpha^\star$, define induced welfare weights

$$\tilde{\pi}_i := \hat{q}_i \alpha_i^\star, \qquad \sum_{i \in V} \tilde{\pi}_i = 1. \qquad (14)$$

We aggregate personalized rewards into a single scalar score

$$\bar{r}_\theta(x, y) = \sum_{i \in V} \tilde{\pi}_i \, r_\theta(x, y, i) = \sum_{i \in V} \hat{q}_i \alpha_i^\star \, r_\theta(x, y, i). \quad (15)$$

When $\hat{q} \approx q$ and $\alpha^\star \approx \pi/q$, we obtain $\tilde{\pi} \approx \pi$, so $\bar{r}_\theta$ approximates the designer-chosen welfare aggregation of individualized utilities.

For large annotator sets, we approximate Eq. (15) via Monte Carlo sampling (Appendix C.4). The same weights $\alpha^\star$ and aggregated reward $\bar{r}_\theta$ can serve as drop-in replacements in downstream DPO/PPO objectives; integration details are in Appendix C.6.

## 4. Theoretical Analysis

We provide theoretical justification for our approach by establishing three key results. First, we show that naive reward modeling implicitly optimizes a $q$-weighted objective (Theorem 4.1), explaining why it fails under biased inclusion. Second, we prove that importance weighting induces an effective mixture $\tilde{\pi} = q \odot \alpha$ (Lemma 4.2), enabling principled correction. Third, we bound the welfare deviation from target $\pi$ by $\|\tilde{\pi} - \pi\|_1$ (Proposition 4.3), quantifying the cost of approximate debiasing. Additional results are in Appendix D.

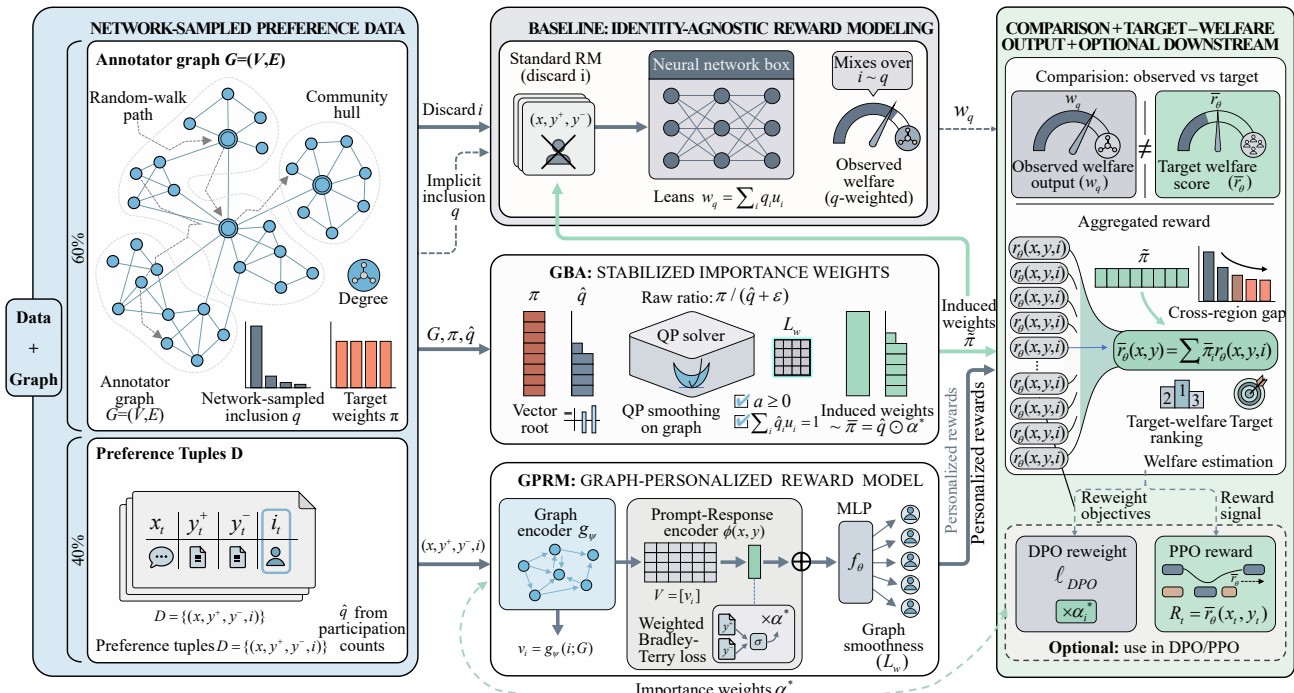

*Figure 2.* **Overview of Graph-Preference Learning. Left**: Network-sampled preference data—annotators form a graph $G = (V, E)$ with biased inclusion $q$ (grey bars) differing from target weights $\pi$ (orange bars); preference tuples $\mathcal{D} = \{(x, y^+, y^-, i)\}$ retain annotator IDs. **Top-middle**: Baseline identity-agnostic reward modeling discards $i$ and learns a $q$-weighted welfare $w_q = \sum_i q_i u_i$. **Bottom-middle**: Our method—GBA solves a QP to compute stabilized weights $\alpha^\star$ and induced welfare $\tilde{\pi} = \hat{q} \odot \alpha^\star$; GPRM learns personalized rewards $r_\theta(x, y, i)$ via graph-based parameter sharing with importance-weighted Bradley–Terry loss. **Right**: Comparison of observed ($w_q$) vs. target-welfare output ($\bar{r}_\theta$); aggregated reward $\bar{r}_\theta(x, y) = \sum_i \tilde{\pi}_i r_\theta(x, y, i)$ targets $\pi$. (Optional) The same weights can reweight DPO/PPO objectives (Appendix).

## 4.1. Naive modeling learns a $q$-mixture objective

We first formalize the basic mixture structure: ignoring annotator identity optimizes a $q$-weighted mixture of annotator-conditional risks.

**Theorem 4.1** (Naive reward modeling minimizes a $q$-mixture population risk). *Let data be generated by sampling an annotator $i \sim q$ and then a comparison $(x, y^+, y^-) \sim \mathcal{T}(i)$, with observed label $y^+ \succ y^-$. Let $r_\theta(x, y)$ be a scalar reward model that ignores $i$, and define*

$$\ell_{\text{naive}}(\theta; x, y^+, y^-) := -\log \sigma\big(r_\theta(x, y^+) - r_\theta(x, y^-)\big).$$

*Then the population risk optimized by maximum-likelihood reward modeling is*

$$\mathcal{L}_{naive}(\theta) = \mathbb{E}_{i \sim q} \mathbb{E}_{(x, y^+, y^-) \sim \mathcal{T}(i)} \big[\ell_{\text{naive}}(\theta; x, y^+, y^-)\big]$$
$$= \sum_{i \in V} q_i \mathcal{L}_i(\theta),$$

$$(16)$$

*where $\mathcal{L}_i(\theta) := \mathbb{E}_{(x, y^+, y^-) \sim \mathcal{T}(i)}[\ell_{\text{naive}}(\theta; x, y^+, y^-)]$. Consequently, whenever gradients interchange with expectation,*

$$\nabla_\theta \mathcal{L}_{naive}(\theta) = \sum_{i \in V} q_i \nabla_\theta \mathcal{L}_i(\theta). \quad (17)$$

*Proof in Appendix D.4.*

Under standard well-specification and welfare-consistency conditions, the learned scalar score is an affine transform of $w_q(x, y) = \sum_i q_i u_i(x, y)$ and induces rankings that optimize the observed welfare; see Appendix D (Assumptions D.1–D.2 and Corollary D.3).

## 4.2. Importance Weighting and Induced Welfare

We now show how importance weighting converts learning under the inclusion distribution $q$ into learning under an induced distribution $\tilde{\pi} = q \odot \alpha$.

**Lemma 4.2** (Importance-weighted objective equals induced mixture). *Let $i \sim q$ and let $\alpha$ be any nonnegative weights satisfying $\sum_{i \in V} q_i \alpha_i = 1$. Define the induced distribution $\tilde{\pi}_i := q_i \alpha_i$. Then for any per-annotator population quantity $h_i(\theta)$,*

$$\sum_{i \in V} q_i \alpha_i h_i(\theta) = \sum_{i \in V} \tilde{\pi}_i h_i(\theta). \quad (18)$$

*In particular, if $q_i > 0$ and $\alpha_i = \pi_i / q_i$, then $\tilde{\pi} = \pi$. Proof in Appendix D.5.*

When $q$ is unknown, we use an estimator $\hat{q}$ in GBA and

obtain induced weights $\tilde{\pi} = \hat{q} \odot \alpha^{\star}$; the same algebra holds with $\hat{q}$ in place of $q$.

### 4.3. Welfare deviation is controlled by weight mismatch

Even when smoothing and other stabilization are active, the welfare loss from using induced weights $\tilde{\pi}$ instead of the target $\pi$ can be bounded by their $\ell_1$ mismatch.

**Proposition 4.3** (Welfare deviation bound via weight mismatch). *Assume $|U_i(\mu_\phi)| \leq R$ for all $i$ and decision rules $\mu_\phi$. Let $\pi$ be the target welfare weights and let $\tilde{\pi}$ be the induced weights (e.g., $\tilde{\pi} = \hat{q} \odot \alpha^{\star}$). Then for any $\mu_\phi$,*

$$\left| W_{\text{target}}(\mu_\phi) - \widetilde{W}(\mu_\phi) \right| \leq R \|\pi - \tilde{\pi}\|_1, \qquad (19)$$

*where $\widetilde{W}(\mu_\phi) := \sum_i \tilde{\pi}_i U_i(\mu_\phi)$. Proof in Appendix D.8.*

**Additional results.** Appendix D provides further results on induced-welfare representation of the aggregated reward (Theorem D.6), community mass control in GBA (Proposition D.7), and finite-sample diagnostics including variance/ESS and decomposition bounds (Proposition D.9, Theorem D.8).

## 5. Experiments

We evaluate Graph-Preference Learning (GPRM+GBA) at the reward/preference-model level. We use (i) synthetic graphs with known individual utilities, enabling direct measurement of target-welfare recovery under biased inclusion, and (ii) a semi-synthetic evaluation on LMArena (Chiang et al., 2024) that uses a session-similarity graph to induce network-sampled inclusion in training and a $\pi$-balanced test set over languages. All evaluations are offline (preference prediction / reranking); we do not run end-to-end policy optimization. On synthetic data, we report welfare estimation error (normalized MSE between a method's public score $s(x, y)$ and the target welfare score $U_\pi(x, y)$), welfare disparity (max–min gap across communities under offline reranking by $s$), and degree bias (correlation between node degree and induced weight mismatch $\tilde{\pi} - \pi$). On LMArena we report pairwise accuracy, pairwise log loss, ECE (calibration), and cross-language accuracy gap (max–min accuracy across languages). Formal definitions are in Appendix F.

**Splits, leakage prevention, and uncertainty reporting. Synthetic:** we split prompts into train/validation/test (by prompt, not by pair) to avoid leakage across candidate outputs, repeat over 10 random seeds, and report mean $\pm$ std (and optionally 95% CIs via bootstrap over seeds). **LMArena:** we use two complementary protocols: (i) prompt-heldout cross-validation (disjoint prompts; sessions may appear across folds) and (ii) session-heldout evaluation as a leakage/robustness check. We report mean $\pm$ std across folds/splits and compute CIs via bootstrap over prompts.

For fairness/gap metrics, we aggregate at the prompt level to avoid overweighting heavy sessions. We construct session graphs from metadata only (no label leakage) and estimate inclusion probabilities $\hat{q}$ from training-fold participation counts with smoothing.

### 5.1. Synthetic Graph Experiment

We evaluate Graph-Preference Learning (GPRM+GBA) on a comprehensive suite of synthetic benchmarks. To ensure robustness, we test across multiple graph topologies: Stochastic Block Models (SBM) for community structure, Barabási–Albert (BA) for scale-free power-law networks, and Watts–Strogatz (WS) for small-world properties. We simulate four distinct feedback-collection (inclusion) mechanisms: Uniform, Degree-biased ($q_i \propto d_i^{1.5}$), Snowball, and Community-biased. We compare against a suite of baselines, including *stabilized IPW* variants: SN-IPW (self-normalized) (Chandak et al., 2021), Clipped-IPW (Chandak et al., 2021), and a QP-stabilized weighting baseline (GBA-QP, weights only)(Hainmueller, 2012), a *personalized no-graph* baseline (Personalized) (Li et al., 2024b), alongside Standard reward modeling (scalar model) (Stiennon et al., 2020; Ouyang et al., 2022), Group-Based reweighting (Chakraborty et al., 2024; Ramesh et al., 2024), GPRM (without GBA), and DRO-CVaR (Rahimian & Mehrotra, 2019). Full details are in Appendix H.

**Main results.** Table 1 summarizes the Welfare-MSE improvement of GPRM+GBA across all graph types and sampling methods. Our method achieves consistent gains (25–41% average), with the largest improvements under community-biased sampling (59–62%), where representation shift is strongest.

*Table 1.* Summary of GPRM+GBA improvement (%) in Welfare MSE across graph types and sampling methods. Higher is better.

| Sampling | SBM | BA | WS |
|---|---|---|---|
| Uniform | 35 | 36 | 24 |
| Degree-Biased | 10 | 36 | 6 |
| Snowball | 18 | 31 | 7 |
| Community-Biased | 59 | 62 | 62 |
| **Average** | **31** | **41** | **25** |

Table 2 presents results on the SBM graph across four inclusion mechanisms. *Uniform sampling* ($q = \pi$): Weighting baselines match Standard, as expected. Personalized achieves the lowest Welfare MSE (48% reduction), while our full method remains competitive (35% reduction). *Degree-biased sampling*: Our method improves Welfare MSE by 10% and substantially reduces degree bias (0.354 vs. 0.900), illustrating effective debiasing. GPRM alone attains better Welfare MSE (16%) but leaves degree bias unchanged, highlighting a debiasing–accuracy tradeoff. *Snow-*

*ball sampling*: Our method achieves the best Welfare MSE (18% reduction). Group-Based yields smallest degree bias but does not improve Welfare MSE, suggesting coarse correction is insufficient. *Community-biased sampling*: Our method delivers the largest gain (59% Welfare-MSE reduction) and reduces degree bias from 0.726 to 0.310, confirming the benefit of targeting $\pi$ under strong representation shifts. Weight-stability diagnostics are in Appendix F.3.

**Cross-graph generalization.** Table 4 shows that our approach generalizes across diverse topologies under degree-biased sampling, improving Welfare MSE by 10% (SBM), 36% (BA), and 6% (WS) while consistently reducing degree bias.

**Ablation.** GPRM alone captures annotator heterogeneity and reduces Welfare MSE, but does not correct representation bias ($q \neq \pi$). Adding GBA yields large gains under strong distribution shift and improves debiasing metrics, though the optimal method can vary by mechanism. Full ablation is in Appendix H.4.

## 5.2. Semi-Synthetic Case Study on LMArena

We present a semi-synthetic case study on the LMArena (Chatbot Arena) dataset (Chiang et al., 2024), which contains over 135,000 pairwise preference comparisons from human evaluators across multiple languages and model categories. LMArena is an open platform where users compare outputs from different LLMs and provide pairwise preferences. We use `evaluation_session_id` as a proxy for annotator identity; after filtering to sessions with at least two interactions, we retain 5,591 sessions for graph construction (Appendix H.2). Because LMArena is not collected via an explicit social network, we treat it as a *pool* of labeled preferences and induce network-sampled inclusion by sub-sampling training feedback using a graph-based sampling procedure. We set a normative target welfare to equal-per-language ($\pi$) and evaluate on a $\pi$-balanced test set.

We construct a *session-similarity* graph $G$ using a fixed-degree $k$NN procedure over session metadata (Appendix H.2). Each session is embedded in a feature space (language, model category, timestamp features); edges connect each node to its $k$ most similar neighbors and are symmetrized. This yields an approximately fixed-degree graph, reducing mechanical degree artifacts. On each prompt-heldout split, we create a biased training set by running a random-walk-with-restart sampling procedure on $G$ (starting from seeds in over-represented languages) and collecting training-fold preference pairs from visited sessions. The induced inclusion distribution $q$ is proportional to visit counts; we estimate $\hat{q}$ from the resulting training data and compute weights $\alpha^\star$. For evaluation, we construct a $\pi$-balanced test set by sampling equal numbers of held-out preference pairs per language.

*Table 2.* Synthetic results on SBM graph (100 nodes, 5 communities). Mean ± std over 10 runs. W-MSE = Welfare MSE; Disp. = Disparity; D-Bias = Degree Bias; Impr. = Improvement vs. Standard. Green = best, Blue = ours. ↓ = lower is better.

| Sampling | Method | W-MSE↓ | Disp.↓ | D-Bias↓ | Impr. |
|---|---|---|---|---|---|
| Uniform | Standard | $0.123_{\pm.08}$ | $3.58_{\pm.77}$ | 0.00 | — |
| | Personalized | $\mathbf{0.064}_{\pm.04}$ | $3.54_{\pm.69}$ | 0.00 | **48%** |
| | IW-Only | $0.114_{\pm.08}$ | $3.52_{\pm.75}$ | 0.00 | 8% |
| | SN-IPW | $0.113_{\pm.06}$ | $3.60_{\pm.75}$ | 0.00 | 9% |
| | Clipped-IPW | $0.125_{\pm.09}$ | $3.53_{\pm.83}$ | 0.00 | −2% |
| | GBA-QP | $0.108_{\pm.07}$ | $3.58_{\pm.73}$ | 0.00 | 12% |
| | Group-Based | $0.242_{\pm.15}$ | $\mathbf{3.50}_{\pm.66}$ | −0.73 | −96% |
| | GPRM (Ours) | $0.088_{\pm.06}$ | $3.60_{\pm.79}$ | 0.00 | 28% |
| | DRO-CVaR | $0.771_{\pm.29}$ | $3.54_{\pm.77}$ | 0.00 | −525% |
| | **GPRM+GBA (Ours)** | $0.080_{\pm.04}$ | $3.56_{\pm.80}$ | **0.00** | 35% |
| Degree | Standard | $0.230_{\pm.17}$ | $3.51_{\pm.76}$ | 0.90 | — |
| | Personalized | $0.210_{\pm.16}$ | $3.51_{\pm.75}$ | 0.90 | 9% |
| | IW-Only | $0.231_{\pm.14}$ | $3.43_{\pm.57}$ | 0.86 | 0% |
| | SN-IPW | $0.284_{\pm.28}$ | $3.52_{\pm.66}$ | 0.86 | −23% |
| | Clipped-IPW | $0.208_{\pm.14}$ | $\mathbf{3.42}_{\pm.66}$ | 0.57 | 10% |
| | GBA-QP | $0.246_{\pm.18}$ | $3.55_{\pm.60}$ | 0.35 | −7% |
| | Group-Based | $0.373_{\pm.24}$ | $3.47_{\pm.60}$ | **0.09** | −62% |
| | GPRM (Ours) | $\mathbf{0.195}_{\pm.15}$ | $3.54_{\pm.77}$ | 0.90 | **16%** |
| | DRO-CVaR | $0.792_{\pm.34}$ | $3.48_{\pm.78}$ | 0.90 | −244% |
| | **GPRM+GBA (Ours)** | $0.207_{\pm.14}$ | $3.51_{\pm.68}$ | 0.35 | 10% |
| Snowball | Standard | $0.150_{\pm.09}$ | $3.53_{\pm.79}$ | 0.88 | — |
| | Personalized | $0.163_{\pm.15}$ | $3.55_{\pm.82}$ | 0.88 | −8% |
| | IW-Only | $0.132_{\pm.08}$ | $3.45_{\pm.77}$ | 0.70 | 12% |
| | SN-IPW | $0.161_{\pm.08}$ | $3.56_{\pm.67}$ | 0.70 | −7% |
| | Clipped-IPW | $0.146_{\pm.07}$ | $3.54_{\pm.82}$ | 0.50 | 3% |
| | GBA-QP | $0.126_{\pm.06}$ | $3.49_{\pm.73}$ | 0.35 | 16% |
| | Group-Based | $0.300_{\pm.18}$ | $3.43_{\pm.70}$ | **0.05** | −100% |
| | GPRM (Ours) | $0.136_{\pm.12}$ | $3.50_{\pm.76}$ | 0.88 | 9% |
| | DRO-CVaR | $0.735_{\pm.38}$ | $\mathbf{3.42}_{\pm.68}$ | 0.88 | −390% |
| | **GPRM+GBA (Ours)** | $\mathbf{0.124}_{\pm.06}$ | $3.49_{\pm.75}$ | 0.35 | **18%** |
| Community | Standard | $0.532_{\pm.44}$ | $3.47_{\pm.75}$ | 0.73 | — |
| | Personalized | $0.502_{\pm.45}$ | $3.46_{\pm.72}$ | 0.73 | 6% |
| | IW-Only | $0.305_{\pm.27}$ | $3.34_{\pm.72}$ | 0.42 | 43% |
| | SN-IPW | $0.329_{\pm.37}$ | $3.39_{\pm.70}$ | 0.42 | 38% |
| | Clipped-IPW | $0.257_{\pm.20}$ | $\mathbf{3.26}_{\pm.64}$ | 0.38 | 52% |
| | GBA-QP | $0.266_{\pm.20}$ | $3.36_{\pm.70}$ | 0.31 | 50% |
| | Group-Based | $0.226_{\pm.17}$ | $3.45_{\pm.75}$ | **0.01** | 57% |
| | GPRM (Ours) | $0.516_{\pm.47}$ | $3.46_{\pm.73}$ | 0.73 | 3% |
| | DRO-CVaR | $0.921_{\pm.45}$ | $3.57_{\pm.86}$ | 0.73 | −73% |
| | **GPRM+GBA (Ours)** | $\mathbf{0.220}_{\pm.14}$ | $3.36_{\pm.75}$ | 0.31 | **59%** |

**Results.** Table 3 reports a *semi-synthetic* LMArena evaluation: we induce biased training inclusion by running a network-sampling procedure on a fixed-degree session similarity graph, then evaluate on a $\pi$-balanced test set (Appendix H.2). Under this setting, GPRM+GBA (ours) achieves 63.6% pairwise accuracy vs. 55.8% for Standard (+18% relative improvement) and reduces the max–min cross-language accuracy gap from 46.0% to 38.4% (17% gap reduction). We also observe substantially improved proper scoring (log loss 0.691 vs. 1.110), supporting the claim that graph-personalized modeling improves reliability under structured inclusion. Notably, GPRM alone (without GBA) improves proper scoring and slightly reduces the cross-language gap relative to Standard (45.3% vs. 46.0%), but loses accuracy, while the full method achieves the best accuracy and the lowest gap (38.4%), highlighting complementary roles of graph-personalized modeling (GPRM) and stabilized reweighting (GBA).

*Table 3.* Semi-synthetic LMArena results (biased training via network sampling on a fixed-degree similarity graph; evaluation on a $\pi$-balanced test set). Accuracy = pairwise ranking accuracy. LogLoss = pairwise negative log-likelihood. ECE = expected calibration error (10 bins). Acc. Gap = max–min accuracy across languages. Err. Red. = relative error reduction vs. Standard. Green = best, Blue = our method.

| Method | Acc.↑ | LogLoss↓ | ECE↓ | Acc. Gap↓ | Err. Red. |
|---|---|---|---|---|---|
| Standard | $55.8_{\pm4.6}\%$ | $1.110_{\pm0.053}$ | 0.323 | 46.0% | — |
| Personalized | $56.9_{\pm1.6}\%$ | $0.692_{\pm0.002}$ | **0.058** | 44.1% | 2% |
| IW-Only | $59.9_{\pm1.7}\%$ | $0.964_{\pm0.025}$ | 0.258 | 58.8% | 9% |
| SN-IPW | $60.0_{\pm2.0}\%$ | $0.990_{\pm0.028}$ | 0.286 | 62.1% | 10% |
| Clipped-IPW | $60.5_{\pm2.1}\%$ | $0.992_{\pm0.018}$ | 0.281 | 62.5% | 11% |
| GBA-QP | $60.0_{\pm1.2}\%$ | $0.983_{\pm0.013}$ | 0.274 | 62.1% | 10% |
| Group-Based | $52.9_{\pm5.1}\%$ | $0.710_{\pm0.005}$ | 0.136 | 54.6% | −7% |
| DRO | $58.9_{\pm2.5}\%$ | $0.988_{\pm0.013}$ | 0.275 | 62.2% | 7% |
| GPRM (Ours) | $54.3_{\pm1.6}\%$ | $0.692_{\pm0.001}$ | 0.069 | 45.3% | −3% |
| **GPRM+GBA (Ours)** | $\mathbf{63.6_{\pm0.8}\%}$ | $\mathbf{0.691_{\pm0.001}}$ | 0.119 | 38.4% | **18%** |

*Table 4.* Cross-graph topology comparison under degree-biased sampling. Green = best, Blue = our method.

| Graph | Method | W-MSE↓ | D-Bias↓ | Impr. |
|---|---|---|---|---|
| SBM | Standard | $0.230_{\pm0.17}$ | 0.900 | — |
| | Personalized | $0.210_{\pm0.16}$ | 0.900 | 9% |
| | GPRM (Ours) | $\mathbf{0.195_{\pm0.15}}$ | 0.900 | **16%** |
| | **GPRM+GBA (Ours)** | $0.207_{\pm0.14}$ | **0.354** | 10% |
| BA | Standard | $0.123_{\pm0.08}$ | 0.914 | — |
| | Personalized | $0.118_{\pm0.09}$ | 0.914 | 4% |
| | GPRM (Ours) | $0.122_{\pm0.09}$ | 0.914 | 1% |
| | **GPRM+GBA (Ours)** | $\mathbf{0.079_{\pm0.05}}$ | **0.771** | **36%** |
| WS | Standard | $0.083_{\pm0.05}$ | 0.994 | — |
| | Personalized | $\mathbf{0.073_{\pm0.04}}$ | 0.994 | **13%** |
| | GPRM (Ours) | $0.090_{\pm0.07}$ | 0.994 | −8% |
| | **GPRM+GBA (Ours)** | $0.078_{\pm0.05}$ | **0.938** | 6% |

**Key findings.** Across synthetic benchmarks and the *semi-synthetic* LMArena case study, our method demonstrates consistent gains when structured inclusion bias is present. On synthetic benchmarks (SBM), it achieves up to 62% welfare-MSE reduction under community-biased sampling and 35% under uniform sampling (Table 2). On semi-synthetic LMArena, it achieves 18% relative accuracy improvement (55.8% to 63.6%) and reduces the cross-language accuracy gap by 17% (from 46.0% to 38.4%), with substantially improved log loss (1.110 to 0.691; Table 3). See Figure 3 for visual comparison of key synthetic metrics.

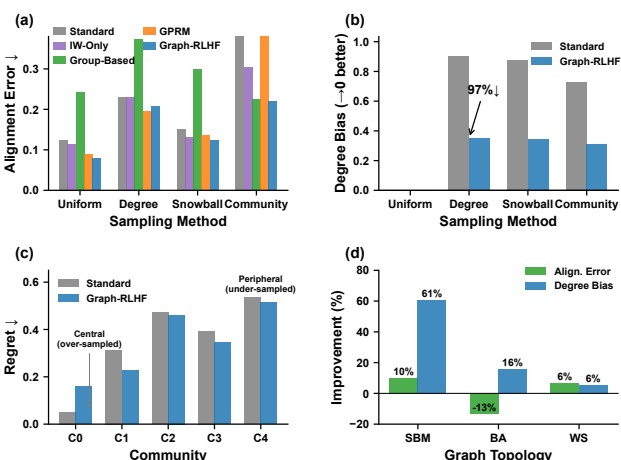

*Figure 3.* Synthetic experiment analysis on SBM graph: (a) Welfare MSE across inclusion mechanisms; (b) Degree-bias correction under biased inclusion; (c) Per-community regret analysis under degree-biased sampling; (d) Cross-graph generalization under degree-biased sampling (Welfare MSE improvement: SBM 10%, WS 6%, BA 36%).

**Robustness and data efficiency.** Our method degrades gracefully under graph noise (Appendix G), maintains stable importance weights under biased inclusion (ESS $\approx$ 0.35–0.51; Appendix F.3), and shows consistent gains across data-sparsity regimes, with the largest relative improvements when feedback is limited (Appendix H.6).

**Ablation insights.** Our ablations (Appendix H.3) reveal three key findings: (1) *Stronger bias $\Rightarrow$ larger gains*: as sampling bias increases ($\gamma$ in $q_i \propto d_i^\gamma$), the gap between Standard and GPRM+GBA widens, confirming that our method provides greater benefit under stronger network-induced bias. (2) *Complementary roles of GPRM and GBA*: GPRM alone captures annotator heterogeneity and improves welfare estimation but does not correct representation bias ($q \neq \pi$); adding GBA yields large additional gains under distribution shift (e.g., 59% under community-biased sampling vs. 3% for GPRM alone; Table 2). (3) *Graph structure matters*: sparse graphs with heterogeneous communities show the largest improvements; in near-uniform-degree graphs (e.g., dense WS), simpler baselines can be competitive, suggesting practitioners should match method complexity to actual inclusion heterogeneity.

# 6. Conclusion and Limitations

We introduced Graph-Preference Learning, a framework for target-welfare preference/reward modeling under network-sampled (or otherwise structured) inclusion. Our approach combines graph-personalized reward modeling with stabilized importance-weighted aggregation, providing theoretical guarantees on welfare deviation.

**Limitations.** (1) We assume the annotator graph is known or can be estimated; in practice, graph construction requires domain expertise. (2) Our theoretical analysis focuses on population-level objectives for reward/preference modeling; end-to-end downstream policy optimization is intentionally out of scope, and we only provide optional integration (DPO/PPO) in the appendix. (3) Our LMArena evaluation is *semi-synthetic*: because LMArena does not include an explicit recruitment/exposure network, we induce network-sampled inclusion by running graph-based sampling on a constructed session-similarity graph and evaluate on a $\pi$-balanced test set. Further evaluation on preference data collected under *genuine* networked recruitment/exposure

mechanisms would strengthen external-validity and generalizability claims. (4) LMArena uses session IDs as annotator proxies rather than true persistent annotator identities; this may underestimate personalization benefits compared to datasets with stable annotator identifiers.

**Practical deployment considerations.** Graph-Preference Learning requires: (i) annotator/session identities linked across tasks, (ii) an annotator graph (constructible from demographics, session metadata, co-rating patterns, or explicit social ties), (iii) inclusion probability estimates, and (iv) a governance choice of target weights $\pi$. When annotator graphs are unavailable, practitioners can use simpler group-based reweighting. When IDs are not persistent, the method reduces to standard preference-based reward modeling with optional group constraints.

## Impact Statement

This paper presents work whose goal is to advance the field of machine learning, specifically addressing fairness and representation in preference-based reward modeling. Our framework enables practitioners to explicitly choose and audit whose preferences are represented in learned reward models, which has implications for democratic governance of AI systems. By making the welfare target $\pi$ an explicit design choice and providing tools to correct for network-induced sampling bias, we aim to promote more equitable and transparent AI alignment. Potential risks include the possibility that poorly chosen target weights could entrench existing biases rather than correct them; we emphasize that defining $\pi$ is a normative choice requiring careful governance. We do not foresee specific negative societal consequences beyond those common to advances in AI alignment research.

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

# A. Related Work

We review work on RLHF and alignment (Section A.1), heterogeneous and personalized preferences (Section A.2), fairness and social choice in reward modeling (Section A.3), structured and graph-based feedback (Section A.4), social network sampling and graph fairness (Section A.5), and LLM-based simulations of human populations (Section A.6). Throughout, we highlight that prior work treats human raters as an i.i.d. population or coarse groups, whereas we explicitly model them as nodes in a graph and correct for network-induced sampling bias.

## A.1. RLHF and Alignment

Modern RLHF pipelines emerged from the InstructGPT work of Ouyang et al. (2022), which popularized the three-stage recipe of supervised fine-tuning, reward modeling from pairwise comparisons, and reinforcement learning to optimize the policy against the learned reward. Subsequent work such as Direct Preference Optimization (DPO) (Rafailov et al., 2023) showed that one can bypass explicit reinforcement learning and directly optimize the policy against preferences, while RLAIF (Lee et al., 2024) demonstrated that AI-generated feedback can partially substitute for human labels to scale alignment. Recent surveys and monographs (Kaufmann et al., 2025; Lambert, 2025; Shen et al., 2023) synthesize this literature, covering reward modeling choices, optimization methods, and open challenges in aligning large language models.

These foundations treat human raters as an unstructured pool: annotations are modeled as i.i.d. draws from an abstract population, sometimes with per-rater reliability or noise parameters, but without any relational structure. The PRISM alignment dataset (Kirk et al., 2024) takes a major step toward richer feedback by collecting participatory, representative, and individualized responses from over 1,500 participants across 75 countries, making explicit the subjectivity and cultural diversity of alignment. More recently, OpenAI's Collective Alignment dataset (OpenAI, 2025) collects value-sensitive preference data from over 1,000 globally distributed annotators with rich demographic information, revealing significant geographic imbalance in recruitment (36% from the US alone). However, even these datasets model participants as a set of individuals with attributes rather than as nodes in a network; their social or institutional ties and resulting sampling patterns are not represented. In contrast, we treat annotators as vertices in a graph and ask how network structure itself shapes the reward signal that RLHF learns.

## A.2. Heterogeneous and Personalized Preferences

A growing line of work studies RLHF under heterogeneous preferences. Poddar et al. (2024) propose variational preference learning, modeling each rater with a latent preference vector so that the model can capture diverse tastes. Personalized RLHF frameworks (Li et al., 2024b) formalize the problem of learning a personalized policy conditioned on user identity, and show that naive global RLHF can perform poorly for minority or atypical users. Parameter-efficient approaches such as shared low-rank adaptations for personalized RLHF (Liu et al., 2025) and reward factorization (Shenfeld et al., 2025) decompose preferences into shared and user-specific components, enabling scalable personalization on top of a base model. A recent survey on personalized alignment (Guan et al., 2025) consolidates these methods and argues that personalization is a missing piece of human-aligned LLMs.

These works acknowledge that "one reward does not fit all" and that per-user conditioning can improve alignment. However, users are still modeled as independent points in a feature space: there is no explicit notion of a social, institutional, or spatial graph connecting them. In particular, they do not address how feedback is collected over networks (e.g., via platforms or peer recruitment), nor how network structure affects which preferences are over-represented. Our work is complementary: we also allow preferences to vary between annotators, but we explicitly embed annotators in a graph and use this structure to regularize per-annotator utilities and to correct for network-induced sampling bias.

## A.3. Fairness, Group Robustness, and Social Choice in RLHF

Several recent papers consider fairness and robustness across groups in RLHF. MaxMin-RLHF (Chakraborty et al., 2024) formulates alignment as a minimax optimization over demographic or preference groups, maximizing the worst-group reward to avoid catastrophic under-alignment. Group Robust Preference Optimization (GRPO) (Ramesh et al., 2024) studies robust preference optimization without explicit reward models, focusing on worst-group regret. Li et al. (2024a) propose fair and stable composition of multiple reward signals in reinforcement learning for language models, controlling disparities and instability between reward components. Ouyang et al. (2025) explicitly analyze reward fairness in RLHF from a resource-allocation perspective, framing the allocation of reward mass across groups as a fairness problem. Complementary

benchmarks such as Song et al. (2025) systematically evaluate group fairness in reward models, revealing disparities across demographic groups even when average performance is high. Work on fairness-aware reward optimization, such as FARO (Choi et al., 2025), further develops constrained objectives and certificates for reward fairness.

Parallel work brings tools from social choice theory into RLHF. Ge et al. (2024) develop axioms for AI alignment from human feedback and show that standard Bradley–Terry models can violate natural social-choice properties, proposing linear social choice rules over pairwise comparisons. Dai & Fleisig (2024) map classic voting rules and impossibility results to RLHF, highlighting how contextual preferences and query selection differ from static elections. Conitzer et al. (2024) argue that social choice should guide AI alignment when dealing with diverse human feedback, advocating for pluralistic alignment objectives and careful choice of welfare functions. More recently, Halpern et al. (2026) introduce pairwise calibrated rewards for pluralistic alignment, designing reward models that better represent multiple value systems simultaneously. From a statistical perspective, Dumoulin et al. (2023) interpret learning from pairwise preferences as density estimation over latent utilities and highlight the misspecification that arises when a single reward is forced to represent heterogeneous annotators.

Taken together, this literature provides both normative desiderata (via social choice) and algorithmic tools (fairness-constrained and group-robust objectives) for aggregating diverse preferences. However, the population of annotators is still treated as an i.i.d. set or partitioned into coarse groups. The structure of interactions among annotators—who is connected, who is central, who influences whom—is absent from the models. Our work extends this agenda by showing that even given a fixed social-welfare target (e.g., equal-per-person or equal-per-community), network-induced sampling can cause RLHF to optimize a very different, degree-weighted objective unless we explicitly correct for it.

### A.4. Structured and Graph-Based Feedback

Beyond human ratings, structured feedback sources are increasingly used in alignment. RLKGF (Yan et al., 2025) proposes reinforcement learning from knowledge-graph feedback, using paths and relations in a knowledge graph as a signal for reward modeling, rather than human annotations. This demonstrates that graph structure can be integrated into RLHF-style pipelines, and that rewards can be derived from structured, relational data. However, in RLKGF the graph nodes are entities and relations in a domain, not human raters. Similarly, RLAIF (Lee et al., 2024) replaces some human preferences with feedback from an auxiliary model, but still treats feedback providers as an unstructured set.

Our setting is different: we focus on graphs of annotators, not graphs of entities or prompts. The primary role of graph structure in our work is to capture how preferences cluster and how feedback is collected and propagated across a social or institutional network. In that sense, we are closer in spirit to graph-aware fairness and sampling work (Section A.5) than to knowledge-graph-based reward shaping.

### A.5. Social Network Sampling and Fairness on Graphs

Social network analysis and graph learning provide rich evidence that network structure profoundly shapes sampling, perception, and fairness. Respondent-driven sampling (RDS) and related methods show that recruiting participants via their social ties leads to biased inclusion probabilities, typically over-sampling high-degree and well-connected individuals; inverse-probability weighting is necessary to obtain unbiased population estimates (Wejnert, 2010). The "majority illusion" of Lerman et al. (2016) demonstrates that behaviors that are globally rare can appear common in local neighborhoods because high-degree nodes disproportionately influence neighbors' perceptions. Antunes et al. (2024) analyze minority representation and ranking under sampling in attributed networks, showing that degree- and centrality-based sampling can distort the apparent prominence of minority groups.

In graph machine learning, structural and attribute bias have been documented and mitigated. EDITS (Dong et al., 2022) introduces a model-agnostic framework to debias graphs for GNNs, addressing both attribute and structural bias. FairSNA (Saxena et al., 2024) surveys algorithmic fairness in social network analysis, covering centrality, influence maximization, and community detection, and emphasizing how biased network structure leads to unfair outcomes. Surveys on fairness-aware GNNs (Chen et al., 2024) and fairness for machine learning on graphs (Laclau et al., 2022) catalog a variety of fairness notions and mitigation strategies for node classification, link prediction, and other graph tasks. Methods such as fairGNN-WOD (Wang et al., 2025) tackle fair graph learning without complete demographic information by leveraging latent structure.

These works collectively argue that graphs are not neutral: who is connected to whom, and who is structurally central or

peripheral, critically shapes both observations and downstream model behavior. Yet, RLHF pipelines to date largely ignore these ideas and treat the set of annotators as if it were sampled uniformly at random from a population. Our work brings insights from network sampling and fair graph learning into RLHF, modeling the rater graph explicitly and using it both to regularize per-annotator rewards and to debias aggregation toward a chosen social welfare target.

### A.6. LLM-Based Simulations and Networked Alignment

Recent work uses LLMs themselves to simulate human populations and social processes. Aher et al. (2023) show that large language models can simulate multiple human participants and replicate classic human-subject experiments, introducing Turing-style experiments as a methodology for prototyping behavioral studies. Liu et al. (2024) train socially aligned language models on simulated social interactions between LLM agents, using multi-agent dialogues to learn social norms and behaviors. A broader survey by Gao et al. (2024) reviews agent-based modeling and simulation with LLM agents, including simulations of social networks, opinion dynamics, and institutional processes.

These works suggest that LLM-based simulations can approximate aspects of human behavior and social structure, and they motivate our use of simulated populations for lightweight experiments. However, the focus is typically on phenomena within the simulated society (e.g., emergent norms), rather than on the sampling and aggregation of human feedback used to train another model. Our framework is complementary: we model how feedback from humans embedded in a network is sampled and aggregated into a reward for RLHF, and we can use LLM-based agents on graphs as a testbed for studying these mechanisms without expensive human studies.

### A.7. Summary

In summary, prior work on RLHF and alignment has developed powerful pipelines for learning from human preferences (Ouyang et al., 2022; Rafailov et al., 2023; Kaufmann et al., 2025; Lambert, 2025; Shen et al., 2023), extended them to heterogeneous and personalized settings (Poddar et al., 2024; Li et al., 2024b; Liu et al., 2025; Shenfeld et al., 2025; Guan et al., 2025), and begun to incorporate fairness and social choice considerations in reward modeling (Chakraborty et al., 2024; Ramesh et al., 2024; Li et al., 2024a; Ouyang et al., 2025; Song et al., 2025; Choi et al., 2025; Ge et al., 2024; Dai & Fleisig, 2024; Conitzer et al., 2024; Halpern et al., 2026; Dumoulin et al., 2023). In parallel, social network analysis and fair graph learning have shown that network structure induces sampling bias, perception distortions, and unfair outcomes (Wejnert, 2010; Lerman et al., 2016; Antunes et al., 2024; Dong et al., 2022; Saxena et al., 2024; Chen et al., 2024; Laclau et al., 2022; Wang et al., 2025). Yet, RLHF methods largely ignore the annotator graph and assume i.i.d. feedback. Our work is, to our knowledge, the first to formalize *preference-based reward modeling* with networked annotators, to characterize the inclusion-weighted welfare implicit in standard scalar reward modeling under graph-based sampling, and to propose a graph-aware personalization and debiasing framework that targets a designer-chosen welfare weighting $\pi$. We validate at the reward-model level (preference prediction, offline reranking, and diagnostic weight stability) and include integration into DPO/PPO pipelines as an optional appendix component.

## B. Additional Details for Problem Formulation

### B.1. Observed inclusion vs. target welfare weights

**Key distinction.**

- **Observed inclusion** $q$**:** who *actually* provides feedback under the collection mechanism.

- **Target weights** $\pi$**:** whose utilities *should* matter (a normative choice).

**Typical targets** $\pi$**:**

- *Equal-per-person:* $\pi_i = 1/|V|$.

- *Equal-per-community:* $\pi_i = 1/(K|C_{k(i)}|)$.

- *Equal-per-language (LMArena case study):* $\pi_i = 1/(L \cdot |L_{\ell(i)}|)$.

In Graph-Preference Learning, we compute induced weights $\tilde{\pi}$ (from $\hat{q}$ and stabilized ratios) to approximate $\pi$ even when $q \neq \pi$.

## B.2. Community Welfare and Disparity

Let $\{C_k\}_{k=1}^K$ be a partition of $V$ into communities (e.g., detected via graph clustering). We define community-level welfare under a policy $\mu_\phi$ as

$$W_k(\mu_\phi) = \frac{1}{|C_k|} \sum_{i \in C_k} U_i(\mu_\phi), \qquad k = 1, \dots, K. \tag{20}$$

We measure community-level disparity via the max–min gap $\max_k W_k(\mu_\phi) - \min_k W_k(\mu_\phi)$, or alternatively the variance $\mathrm{Var}_k[W_k(\mu_\phi)]$. When $q_i \propto d_i$ (degree-proportional sampling), communities with higher average degree contribute disproportionately to the learned reward, potentially causing large welfare gaps across communities.

## B.3. Network-Based Sampling and Inclusion Probabilities

In many practical data-collection pipelines, annotators are recruited or exposed to feedback interfaces through mechanisms that can be approximated by network-based sampling. Two canonical examples are:

- **Random walks on $G$.** Suppose at each step we select a node $i$ and then move to a random neighbor $j \in N(i)$, and solicit feedback from the current node. Under mild conditions, the stationary distribution of this Markov chain is proportional to node degree: $q_i \propto d_i$.

- **Respondent-driven sampling.** In respondent-driven sampling (RDS), initial "seed" participants recruit neighbors (e.g., friends or colleagues), who in turn recruit further participants. The resulting sample is known to over-represent high-degree nodes; estimators of the inclusion probabilities $q_i$ often take the form $q_i \propto d_i$ or functions thereof.

These mechanisms are not meant to be exact models of how RLHF data is collected, but they capture the qualitative fact that annotators with larger degree or centrality in $G$ tend to have higher $q_i$. Our method does not require knowing the exact sampling mechanism; Section 3 works with any estimator $\hat{q}_i$ of the inclusion probabilities.

## B.4. Consistency of Scalar Reward Modeling under the Population Model

**Assumption B.1** (Scalar population reward under $q$). There exists a scalar function $r_q^\star(x, y)$ such that for all $x, y^+, y^-$,

$$\Pr(y^+ \succ y^- \mid x) = \sigma(r_q^\star(x, y^+) - r_q^\star(x, y^-)), \tag{21}$$

where the probability is taken over the data-generating process that samples $i \sim q$ and then draws a comparison from $\mathcal{T}(i)$.

For completeness, we sketch why standard scalar reward modeling recovers the $q$-weighted social welfare under Assumption B.1.

Under Assumption B.1, there exists a scalar reward $r_q^\star(x, y)$ such that the probability that a randomly observed preference pair satisfies $y^+ \succ y^-$ is

$$\Pr(y^+ \succ y^- \mid x) = \sigma(r_q^\star(x, y^+) - r_q^\star(x, y^-)). \tag{22}$$

When individual preferences follow a logistic model $\Pr(y^+ \succ y^- \mid x, i) = \sigma(u_i(x, y^+) - u_i(x, y^-))$, the marginal is a mixture: $\Pr(y^+ \succ y^- \mid x) = \sum_i q_i \sigma(\Delta u_i)$. If individual utilities are not too dispersed, this mixture is well-approximated by a single logistic model with logit

$$\ell(x, y^+, y^-) \approx \sum_{i \in V} q_i \left( u_i(x, y^+) - u_i(x, y^-) \right), \tag{23}$$

and $r_q^\star(x, y) \propto \sum_i q_i u_i(x, y)$.

Training a scalar reward model $r_\theta(x, y)$ by maximum-likelihood with the standard Bradley–Terry loss,

$$\mathcal{L}_{\mathrm{BTL}}(\theta) = -\sum_{t=1}^T \log \sigma\left(r_\theta(x_t, y_t^+) - r_\theta(x_t, y_t^-)\right), \tag{24}$$

yields a consistent estimator of $r_q^\star$ as $T \to \infty$ under standard regularity conditions. Consequently, if this scalar reward is used in downstream optimization (e.g., PPO-style RLHF), the downstream procedure is effectively maximizing the $q$-weighted social welfare

$$W_{\text{obs}}(\mu_\phi) = \sum_{i \in V} q_i \, U_i(\mu_\phi), \tag{25}$$

not the target welfare $W_{\text{target}}$ in Definition 2.4. This mismatch is precisely the sampling-induced welfare gap quantified in (8).

## C. Additional Details for Method

**Scope note.** The core contribution of this paper is target-welfare preference/reward modeling (GPRM+GBA) and the aggregated score $\bar{r}_\theta$. This appendix section provides implementation details (e.g., solving for $\alpha^\star$) and optional extensions/integration recipes (PPO/DPO, active querying) for completeness; these optional pieces are not evaluated in our experiments.

### C.1. Optional GPRM details: rater scale and heterophily

**Annotator-specific scale.** To model rater strictness, we can introduce an annotator-specific scale

$$s_i = \exp(g_\xi(v_i)), \tag{26}$$

with a small network $g_\xi$. When used, $s_{i_t}$ multiplies the preference logit in Eq. (11), e.g., $\log \sigma\big(s_{i_t}[r_\theta(x_t, y_t^+, i_t) - r_\theta(x_t, y_t^-, i_t)]\big)$, and one may regularize scales via a term such as $\lambda_s \sum_i \pi_i (\log s_i)^2$. If unused, we set $s_i \equiv 1$.

**Heterophily and over-smoothing.** The Laplacian term implicitly assumes some degree of homophily. In strongly heterophilous graphs, large $\lambda_v$ can over-smooth genuine preference differences. In practice we recommend: (i) tuning $\lambda_v$ conservatively; (ii) constructing $G$ using similarity edges (e.g., co-rating agreement or attribute similarity) rather than arbitrary proximity; and (iii) downweighting edges that connect annotators with systematically divergent preferences.

### C.2. Estimating inclusion probabilities

We estimate $\hat{q}_i$ using one of: **(i) Empirical counts (preferred):** $\hat{q}_i = (n_i + \tau)/\sum_j (n_j + \tau)$, where $n_i$ is annotator $i$'s participation count and $\tau > 0$ smooths rare annotators (ensuring $\hat{q}_i > 0$). **(ii) Degree-/centrality-based:** $\hat{q}_i \propto d_i^\gamma$ when only $G$ is available (we ablate $\gamma$). **(iii) Hybrid:** combine counts and centrality when both are available but noisy.

### C.3. Solving the Graph-Balanced Weight Optimization

Recall the optimization problem for the importance weights $\alpha^\star$:

$$\min_{\alpha \in \mathbb{R}^{|V|}} \quad \frac{1}{2}\|\alpha - \tilde{\alpha}\|_2^2 + \lambda_{\text{smooth}}\, \alpha^\top L_w \alpha + \lambda_{\text{comm}} \sum_{k=1}^{K} \Big( \sum_{i \in C_k} \hat{q}_i \alpha_i - \rho_k \Big)^2 + \lambda_{\text{var}} \sum_{i \in V} \hat{q}_i \alpha_i^2$$
$$\text{s.t.} \quad \alpha_i \geq 0 \; \forall i \in V, \quad \sum_{i \in V} \hat{q}_i \alpha_i = 1, \tag{27}$$

where $\tilde{\alpha}_i = \pi_i/(\hat{q}_i + \varepsilon)$ as defined in (12), $L_w$ is the weighted graph Laplacian, $\lambda_{\text{smooth}} \geq 0$ controls graph smoothness, $\lambda_{\text{comm}} \geq 0$ controls community balance, $\lambda_{\text{var}} \geq 0$ controls variance (and hence ESS), and $\rho_k$ are the target community masses in the induced distribution (e.g., $\rho_k = 1/K$ for equal-per-community welfare).

Problem (27) is a convex quadratic program (QP). In our experiments, we solve it using a projected gradient method:

1. Initialize $\alpha^{(0)} = \tilde{\alpha}$, then project onto the feasible set $\{\alpha \geq 0, \sum_i \hat{q}_i \alpha_i = 1\}$.

2. At iteration $s$, compute the gradient. Let $A_k(\alpha) = \sum_{i \in C_k} \hat{q}_i \alpha_i$ denote the induced community-$k$ mass. Then

$$\nabla f(\alpha^{(s)})_i = \alpha_i^{(s)} - \tilde{\alpha}_i + 2\lambda_{\text{smooth}}(L_w \alpha^{(s)})_i + 2\lambda_{\text{comm}} \hat{q}_i (A_{k(i)}(\alpha^{(s)}) - \rho_{k(i)}) + 2\lambda_{\text{var}} \hat{q}_i \alpha_i^{(s)}, \tag{28}$$

where $k(i)$ is the community containing annotator $i$.

3. Take a gradient step $z^{(s+1)} = \alpha^{(s)} - \eta \nabla f(\alpha^{(s)})$ with step size $\eta$.

4. Project $z^{(s+1)}$ onto the feasible set $\{\alpha \geq 0, \sum_i \hat{q}_i \alpha_i = 1\}$ using weighted simplex projection.

**Weighted simplex projection.** We project onto the convex set $\mathcal{F} := \{\alpha \in \mathbb{R}^{|V|} : \alpha \geq 0, \sum_i \hat{q}_i \alpha_i = 1\}$, assuming $\hat{q}_i > 0$ (ensured by smoothing). The Euclidean projection of a vector $z \in \mathbb{R}^{|V|}$ onto $\mathcal{F}$ has the KKT form

$$\Pi_{\mathcal{F}}(z)_i = \max\{0, \ z_i - \lambda \hat{q}_i\},$$

where $\lambda$ is chosen so that $\sum_i \hat{q}_i \Pi_{\mathcal{F}}(z)_i = 1$. We compute $\lambda$ by a sorting-based threshold search over breakpoints $z_i/\hat{q}_i$ (equivalently, by 1D bisection), yielding an $O(|V| \log |V|)$ projection.

The Laplacian term $\alpha^\top L_w \alpha$ can be computed efficiently using sparse matrix–vector operations. In practice we find that a small number of iterations (e.g., 50–100) suffices for convergence on graphs with up to tens of thousands of annotators.

**Proposition C.1** (Stability of $\alpha^\star$ under perturbations). *Let $J(\alpha)$ be the objective in* (13) *(ignoring constraints for clarity). Its Hessian is*

$$H = I + 2\lambda_{\text{smooth}} L_w + 2\lambda_{\text{comm}} B^\top B + 2\lambda_{\text{var}} \text{diag}(\hat{q}),$$

*where $B$ encodes community-mass linear forms. Since $L_w \succeq 0$, $B^\top B \succeq 0$, and $\text{diag}(\hat{q}) \succeq \hat{q}_{\min} I$, we have $H \succeq \mu I$ with $\mu = 1 + 2\lambda_{\text{var}} \hat{q}_{\min}$. Therefore the unconstrained minimizer mapping is $(1/\mu)$-Lipschitz in the gradient, and the constrained minimizer over a convex feasible set remains nonexpansive under projection. In particular, small perturbations in $(L_w, \hat{q}, \tilde{\alpha})$ imply $\|\alpha^\star - \alpha^{\star\prime}\|_2 \leq \frac{1}{\mu} \|\nabla J - \nabla J'\|_2$.*

### C.4. Monte Carlo approximation of the aggregated score

When $|V|$ is large, computing $\bar{r}_\theta(x,y) = \sum_{i \in V} \tilde{\pi}_i r_\theta(x,y,i)$ exactly can be expensive. We can estimate it via Monte Carlo: sample $i_1, \ldots, i_M \overset{\text{i.i.d.}}{\sim} \tilde{\pi}$ and use

$$\hat{r}_\theta^{\text{MC}}(x,y) := \frac{1}{M} \sum_{m=1}^{M} r_\theta(x,y,i_m),$$

which is unbiased and has variance decreasing as $O(1/M)$.

---

**Algorithm 1** Optional: Monte Carlo aggregation for $\bar{r}_\theta(x,y)$

---

**Require:** Input-output pair $(x,y)$, personalized reward model $r_\theta(x,y,i)$, induced weights $\tilde{\pi}$ over $V$, sample size $M$.
  1: Sample indices $i_1, \ldots, i_M \sim \tilde{\pi}$ (with replacement).
  2: Return $\hat{r}_\theta^{\text{MC}}(x,y) \leftarrow \frac{1}{M} \sum_{m=1}^{M} r_\theta(x,y,i_m)$.

---

### C.5. Approximation to Target Welfare

We justify why using the induced welfare weights $\tilde{\pi}_i = \hat{q}_i \alpha_i^\star$ in the reward aggregation approximates optimizing the target welfare $W_{\text{target}}(\mu_\phi)$.

Assume that: (i) the personalized reward model $r_\theta(x,y,i)$ is a well-calibrated estimator of the latent utility $u_i(x,y)$ (in the sense that $r_\theta(x,y,i) \approx u_i(x,y)$ up to a global affine transformation), and (ii) the importance weights $\alpha_i^\star$ solve the debiasing problem in Section 3.2, so that the induced distribution $\tilde{\pi}_i = \hat{q}_i \alpha_i^\star$ approximates $\pi_i$.

Given a decision rule $\mu_\phi$, the expected aggregated reward is

$$\mathbb{E}_{x,y}[\bar{r}_\theta(x,y)] = \mathbb{E}_{x \sim p_{\text{data}}, y \sim \mu_\phi(\cdot|x)} \left[ \sum_{i \in V} \tilde{\pi}_i \, r_\theta(x,y,i) \right]$$

$$\approx \sum_{i \in V} \tilde{\pi}_i \, \mathbb{E}_{x,y}[u_i(x,y)] = \sum_{i \in V} \tilde{\pi}_i \, U_i(\mu_\phi) = \widetilde{W}(\mu_\phi). \tag{29}$$

By Proposition 4.3, the difference between $\widetilde{W}(\mu_\phi)$ and $W_{\text{target}}(\mu_\phi)$ is bounded by $R\|\pi - \tilde{\pi}\|_1$, where $R$ is the utility range. When $\alpha_i^\star = \pi_i/q_i$ exactly (no smoothing or community constraints), we have $\tilde{\pi} = \pi$ and the approximation is exact.

### C.6. Optional downstream integration: PPO with aggregated reward $\bar{r}_\theta$

As an optional integration (not evaluated in our experiments), we can leverage the aggregated reward $\bar{r}_\theta(x, y)$ directly in a reward-model-based RLHF setup. We outline a PPO variant here for completeness.

At training time, the policy $\mu_\phi$ interacts with an environment (or generates trajectories from a dataset of prompts) to produce sequences $\tau = (x_1, y_1, \ldots, x_T, y_T)$. For each time step $t$, we compute a scalar reward

$$R_t = \bar{r}_\theta(x_t, y_t), \tag{30}$$

where $\bar{r}_\theta$ is defined in (15). We then compute advantages $A_t$ using a standard value-function baseline and optimize the PPO objective

$$\mathcal{L}_{\text{PPO}}(\phi) = -\mathbb{E}_t \left[ \min \left( r_t(\phi) A_t, \, \text{clip}\left( r_t(\phi), 1 - \epsilon, 1 + \epsilon \right) A_t \right) \right], \tag{31}$$

where $r_t(\phi) = \mu_\phi(y_t \mid x_t) / \mu_{\phi_{\text{old}}}(y_t \mid x_t)$ and $\epsilon$ is the clipping parameter.

Because $\bar{r}_\theta$ incorporates the induced welfare weights $\tilde{\pi}_i = \hat{q}_i \alpha_i^\star$, the resulting policy is encouraged to maximize the induced welfare $\widetilde{W}(\mu_\phi)$ rather than the observed welfare $W_{\text{obs}}(\mu_\phi)$ that would be induced by a reward model trained without importance weighting. When importance weights achieve exact debiasing ($\alpha_i^\star = \pi_i / q_i$), the induced welfare equals the target welfare.

### C.7. Optional extension: Graph-Active Feedback Selection

Algorithm 2 describes an optional data-collection extension (not used in our experiments) that selects which annotators to query to improve coverage and reduce representation mismatch.

---

**Algorithm 2** Graph-Active Feedback Selection

---

**Require:** Annotator graph $G = (V, E)$, target weights $\pi_i$, current induced weights $\tilde{\pi}_i = \hat{q}_i \alpha_i$, personalized rewards $r_\theta$, budget $B$.

1: Sample a batch of prompts $\{x_s\}_{s=1}^S$ and candidate outputs.
2: **for** each annotator $i \in V$ **do**
3:     Compute a disagreement score

$$\text{disc}(i) = \frac{1}{S} \sum_{s=1}^S \left| r_\theta(x_s, y_s^\star, i) - \bar{r}_\theta(x_s, y_s^\star) \right|,$$

    where $y_s^\star$ is a reference output (e.g., sampled from the current policy).
4:     Compute an under-representation score $\text{repr}(i) = \pi_i / (\tilde{\pi}_i + \epsilon)$.
5:     Set $a(i) = \alpha_{\text{disc}} \cdot \text{disc}(i) + \alpha_{\text{repr}} \cdot \text{repr}(i)$.
6: **end for**
7: Select a batch $S \subset V$ of size $B$ by greedily adding the node with largest $a(i)$ minus a redundancy penalty for neighbors already in $S$:

$$i^\star = \arg \max_{i \in V \setminus S} \left( a(i) - \alpha_{\text{red}} \sum_{j \in S} \mathbb{I}\{(i, j) \in E\} \right).$$

8: Query annotators in $S$ for new preference labels and update the GPRM and importance weights $\alpha_i$ (hence $\tilde{\pi}_i = \hat{q}_i \alpha_i$).

---

The redundancy penalty discourages selecting annotators that are immediate neighbors of each other, encouraging coverage of diverse regions of the graph. The hyperparameters $\alpha_{\text{disc}}, \alpha_{\text{repr}}, \alpha_{\text{red}}$ control the trade-off between disagreement, under-representation, and exploration of the graph.

## D. Additional Theoretical Details

### D.1. Additional statements referenced from the main text

#### D.1.1. FROM MIXTURE RISK TO $q$-WEIGHTED WELFARE

To connect Theorem 4.1 to welfare, we adopt standard well-specification and welfare-consistency assumptions.

**Assumption D.1** (Well-specified scalar reward model). There exists a scalar reward function $r^\star(x, y)$ such that for all $x, y^+, y^-$,

$$\Pr(y^+ \succ y^- \mid x) = \sigma\big(r^\star(x, y^+) - r^\star(x, y^-)\big), \tag{32}$$

and $r^\star$ is uniquely identified up to an additive constant by minimizing $\mathcal{L}_{\text{naive}}$.

**Assumption D.2** (Welfare-consistent scalar reward under $q$). There exist constants $a > 0$ and $b \in \mathbb{R}$ such that for all $(x, y)$,

$$r^\star(x, y) = a\, w_q(x, y) + b \qquad \text{where} \qquad w_q(x, y) := \sum_{i \in V} q_i\, u_i(x, y). \tag{33}$$

Equivalently, maximizing $\mathbb{E}_{x, y \sim \mu_\phi}[r^\star(x, y)]$ is equivalent to maximizing the observed welfare $W_{\text{obs}}(\mu_\phi) = \sum_{i \in V} q_i U_i(\mu_\phi)$.

**Corollary D.3** ($q$-weighted welfare under welfare-consistency). *Assume Assumptions D.1 and D.2 hold so that naive reward modeling recovers $r^\star$ (up to an additive constant) and $r^\star$ is an affine transform of $w_q$. Then, for any decision rule $\mu_\phi$, maximizing $\mathbb{E}_{x, y \sim \mu_\phi}[r^\star(x, y)]$ is equivalent to maximizing $W_{\text{obs}}(\mu_\phi) = \sum_{i \in V} q_i U_i(\mu_\phi)$. In particular, for any fixed prompt $x$ and finite candidate set $\mathcal{Y}_x^{\text{cand}}$, the ranking induced by $r^\star(x, \cdot)$ coincides with the ranking induced by $w_q(x, \cdot)$. Proof in Appendix D.4.*

Corollary D.3 makes precise the key issue: when $q$ correlates with graph centrality (e.g., $q_i \propto d_i^\gamma$), naive preference learning represents a degree-skewed welfare $w_q$ rather than the normative target $w_\pi$.

### D.1.2. ASSUMPTIONS AND INDUCED-WELFARE REPRESENTATION

When $q$ is unknown, GBA uses an estimator $\hat{q}$ and produces induced weights $\tilde{\pi} = \hat{q} \odot \alpha^\star$.

**Assumption D.4** (Consistent inclusion-probability estimates). Let $\hat{q}_i$ be the estimator of $q_i$ used to construct $\alpha$. Assume $\hat{q}_i \to q_i$ almost surely as $T \to \infty$ for all $i \in V$, and $\min_i q_i > 0$.

**Assumption D.5** (Expressive and calibratable personalized reward model). The function class $\{r_\theta(x, y, i)\}$ is rich enough that there exist parameters $\theta^\star$ and constants $a > 0$, $b \in \mathbb{R}$ such that

$$r_{\theta^\star}(x, y, i) = a\, u_i(x, y) + b \qquad \text{for all } (x, y, i) \text{ in the support of the data.} \tag{34}$$

**Theorem D.6** (Aggregated reward represents induced welfare). *Assume (i) $\hat{q} = q$ and $\alpha^\star$ satisfies $\sum_i q_i \alpha_i^\star = 1$, (ii) Assumption D.5 holds, and (iii) the aggregated reward is constructed as*

$$\bar{r}_{\theta^\star}(x, y) = \sum_{i \in V} \tilde{\pi}_i\, r_{\theta^\star}(x, y, i), \qquad \tilde{\pi}_i := q_i \alpha_i^\star.$$

*Then $\bar{r}_{\theta^\star}$ is an affine transform of the induced welfare score:*

$$\bar{r}_{\theta^\star}(x, y) = a \sum_{i \in V} \tilde{\pi}_i\, u_i(x, y) + b = a\, w_{\tilde{\pi}}(x, y) + b, \tag{35}$$

*where $w_{\tilde{\pi}}(x, y) := \sum_i \tilde{\pi}_i u_i(x, y)$. Moreover, under exact debiasing $\alpha_i^\star = \pi_i / q_i$, we have $\tilde{\pi} = \pi$ and hence $\bar{r}_{\theta^\star}(x, y) = a\, w_\pi(x, y) + b$. Proof in Appendix D.6.*

### D.1.3. COMMUNITY CONTROL AND FINITE-SAMPLE DIAGNOSTICS

**Proposition D.7** (Community mass control under GBA). *Let $\alpha^\star$ solve Eq. (13) and define $A_k(\alpha) = \sum_{i \in C_k} \hat{q}_i \alpha_i$. For $\lambda_{\text{comm}} > 0$, the problem has a unique solution. Moreover, as $\lambda_{\text{comm}} \to \infty$, we have $A_k(\alpha^\star) \to \rho_k$ for all $k$. Proof in Appendix D.7.*

**Theorem D.8** (Welfare deviation decomposition). *Assume $|U_i(\mu_\phi)| \le R$ for all $i$ and decision rules $\mu_\phi$. Let $\pi$ be the target welfare weights and let $\tilde{\pi} = \hat{q} \odot \alpha^\star$ be the induced weights produced by GBA. Suppose there exist parameters $\theta^\star$ and constants $a > 0$, $b \in \mathbb{R}$ such that*

$$\sup_{x, y, i} \big|r_{\theta^\star}(x, y, i) - (a\, u_i(x, y) + b)\big| \le \varepsilon_{\text{RM}},$$

*and the learned model $r_\theta$ (trained from $T$ weighted samples) satisfies*

$$\sup_{x,y,i} \left| r_\theta(x,y,i) - r_{\theta^\star}(x,y,i) \right| \leq \varepsilon_{\mathrm{stat}}(T, \alpha^\star).$$

*Define the (normalized) welfare proxy represented by the learned aggregated reward*

$$\widehat{W}_\theta(\mu_\phi) := \frac{1}{a} \Big( \mathbb{E}_{x \sim p_{data}, \, y \sim \mu_\phi(\cdot|x)}[\bar{r}_\theta(x,y)] - b \Big),$$

*where $\bar{r}_\theta$ is defined in Eq. (15). Then for any $\mu_\phi$,*

$$\left| W_{\mathrm{target}}(\mu_\phi) - \widehat{W}_\theta(\mu_\phi) \right| \leq R\|\pi - \tilde{\pi}\|_1 + \varepsilon_{\mathrm{RM}} + \varepsilon_{\mathrm{stat}}(T, \alpha^\star), \tag{36}$$

*where $\varepsilon_{\mathrm{stat}}$ captures finite-sample effects and decreases as the effective sample size of the weighted data increases. Proof in Appendix D.10.*

## D.2. Finite-Sample Effects: Weight Variance and Effective Sample Size

Importance weighting can dramatically increase variance when some $q_i$ are small. This motivates the explicit variance/ESS regularization term in GBA.

Define the (population) second moment of weights under $q$:

$$m_2(\alpha; q) := \sum_{i \in V} q_i \alpha_i^2. \tag{37}$$

When $\sum_i q_i \alpha_i = 1$, a common notion of population effective sample size factor is $\mathrm{ESS}_{\mathrm{pop}}(\alpha; q) := 1/m_2(\alpha; q) \in (0, 1]$.

**Proposition D.9** (Variance cost of importance weighting). *Let $Z_t \sim q$ i.i.d. and let $H(Z_t)$ be any random variable with $|H(Z_t)| \leq R$ almost surely. Consider the importance-weighted estimator*

$$\widehat{\mu} = \frac{1}{T} \sum_{t=1}^T \alpha_{Z_t} H(Z_t) \qquad of \qquad \mu = \sum_{i \in V} q_i \alpha_i \, \mathbb{E}[H(Z) \mid Z = i] = \sum_{i \in V} \tilde{\pi}_i \, \mathbb{E}[H(Z) \mid Z = i].$$

*If $\sum_i q_i \alpha_i = 1$, then*

$$\mathrm{Var}(\widehat{\mu}) \leq \frac{R^2}{T} m_2(\alpha; q) = \frac{R^2}{T \, \mathrm{ESS}_{\mathrm{pop}}(\alpha; q)}. \tag{38}$$

*In particular, exact debiasing $\alpha_i = \pi_i/q_i$ yields $m_2(\alpha; q) = \sum_i \pi_i^2/q_i$, which can be large when some $q_i$ are small. Proof in Appendix D.9.*

Proposition D.9 motivates the GBA variance penalty $\lambda_{\mathrm{var}} \sum_i \hat{q}_i \alpha_i^2$, which directly controls the empirical analogue of $m_2(\alpha; q)$ (and hence increases effective sample size). In practice, additional stabilization such as clipping or self-normalized weights can further control this variance; we view these as complementary techniques. Accordingly, we include stabilized IPW baselines (SN-IPW, Clipped-IPW) and a QP-stabilized weighting baseline (GBA-QP) in experiments, and report ESS and weight-mismatch diagnostics to make the variance–bias tradeoff auditable.

**Implication for downstream optimization (optional).** While our main contribution and evaluation are at the reward-model level, the same induced-mixture algebra implies that if a downstream RLHF/DPO procedure optimizes an objective built from $\bar{r}_\theta$ or reweights preferences by $\alpha^\star$, it will optimize an induced welfare with weights $\tilde{\pi}$. Formal statements for Graph-weighted DPO/RLHF are provided in the appendix.

## D.3. Mixture-of-Logits Approximation Bound

This section provides the formal bound mentioned in Section 2. Suppose individual preferences follow a logistic model $\Pr(y^+ \succ y^- \mid x, i) = \sigma(\Delta u_i)$ where $\Delta u_i := u_i(x, y^+) - u_i(x, y^-)$. The marginal preference probability is a mixture of logits: $\Pr(y^+ \succ y^- \mid x) = \sum_{i \in V} q_i \sigma(\Delta u_i)$. Let $m := \mathbb{E}_{i \sim q}[\Delta u_i]$ and let $\mathrm{range}(\Delta u) := \max_i \Delta u_i - \min_i \Delta u_i$. Using that $\sigma$ is $1/4$-Lipschitz, we have the approximation bound

$$\left| \mathbb{E}_{i \sim q}[\sigma(\Delta u_i)] - \sigma(m) \right| \leq \frac{1}{4} \mathbb{E}_{i \sim q} |\Delta u_i - m| \leq \frac{1}{8} \mathrm{range}(\Delta u). \tag{39}$$

Thus when preference heterogeneity is small on a given comparison, a scalar logit $\sigma(m)$ approximates the mixture well.

*Proof.* Let $X$ be the random variable $X = \Delta u_i$ when $i \sim q$, and let $m = \mathbb{E}[X]$. Since the derivative of the sigmoid satisfies $\sup_z \sigma'(z) = 1/4$, the function $\sigma(\cdot)$ is $1/4$-Lipschitz: $|\sigma(a) - \sigma(b)| \leq \frac{1}{4}|a - b|$ for all $a, b \in \mathbb{R}$. Therefore,

$$\left| \mathbb{E}[\sigma(X)] - \sigma(m) \right| = \left| \mathbb{E}[\sigma(X)] - \mathbb{E}[\sigma(m)] \right| \leq \mathbb{E}\big[|\sigma(X) - \sigma(m)|\big] \leq \frac{1}{4}\,\mathbb{E}[|X - m|].$$

For the second inequality in Eq. (39), note that $X$ is supported on the interval $[\min_i \Delta u_i, \max_i \Delta u_i]$ and the mean $m$ lies in the same interval. A standard extremal argument for bounded random variables gives $\mathbb{E}|X - m| \leq \frac{1}{2}(\max_i \Delta u_i - \min_i \Delta u_i)$, yielding $\mathbb{E}|X - m| \leq \frac{1}{2}\mathrm{range}(\Delta u)$ and hence Eq. (39). $\qquad\square$

### D.4. Proofs for Theorem 4.1 and Corollary D.3

#### D.4.1. PROOF OF THEOREM 4.1

*Proof.* Let $(I, X, Y^+, Y^-)$ be drawn by first sampling $I \sim q$ over $V$ and then $(X, Y^+, Y^-) \sim \mathcal{T}(I)$. By the law of total expectation,

$$\mathcal{L}_{\mathrm{naive}}(\theta) = \mathbb{E}\big[\ell_{\mathrm{naive}}(\theta; X, Y^+, Y^-)\big] = \sum_{i \in V} \Pr(I = i)\,\mathbb{E}\big[\ell_{\mathrm{naive}}(\theta; X, Y^+, Y^-) \mid I = i\big],$$

which is exactly (16), with $\mathcal{L}_i(\theta) = \mathbb{E}_{(x,y^+,y^-)\sim\mathcal{T}(i)}[\ell_{\mathrm{naive}}(\theta; x, y^+, y^-)]$.

For the gradient identity, assume $\nabla_\theta \ell_{\mathrm{naive}}(\theta; \cdot)$ is dominated by an integrable function in a neighborhood of $\theta$ (or simply bounded, which holds if rewards are clipped / regularized and the dataset support is bounded). Then differentiation under the integral is valid, yielding

$$\nabla_\theta \mathcal{L}_{\mathrm{naive}}(\theta) = \nabla_\theta \sum_{i \in V} q_i \mathcal{L}_i(\theta) = \sum_{i \in V} q_i \nabla_\theta \mathcal{L}_i(\theta),$$

which is (17). $\qquad\square$

#### D.4.2. PROOF OF COROLLARY D.3

*Proof.* **Step 1 (consistency of naive reward modeling).** Under Assumption D.1, there exists a scalar reward function $r^\star$ such that for all $(x, y^+, y^-)$,

$$\Pr(y^+ \succ y^- \mid x) = \sigma\big(r^\star(x, y^+) - r^\star(x, y^-)\big),$$

where the probability is with respect to the marginal data-generating process (induced by sampling $i \sim q$ and then $(x, y^+, y^-) \sim \mathcal{T}(i)$). Training $r_\theta(x, y)$ by maximum likelihood with Bradley–Terry loss is equivalent to logistic regression on pairwise differences, and the negative log-likelihood is minimized (over a rich enough function class) when the model probabilities match the true conditional probabilities. Since the sigmoid link is strictly monotone, this identifies $r^\star$ up to an additive constant. Thus, with sufficient data and capacity, naive reward modeling recovers $r^\star$ (up to a constant).

**Step 2 (reward maximization equals $q$-weighted welfare).** Assumption D.2 states that there exist constants $a > 0$ and $b \in \mathbb{R}$ such that

$$r^\star(x, y) = a \sum_{i \in V} q_i u_i(x, y) + b.$$

Therefore, for any decision rule $\mu_\phi$,

$$\mathbb{E}_{x \sim p_{\mathrm{data}},\, y \sim \mu_\phi(\cdot|x)}[r^\star(x, y)] = a\,\mathbb{E}_{x,y}\left[\sum_{i \in V} q_i u_i(x, y)\right] + b = a \sum_{i \in V} q_i U_i(\mu_\phi) + b = a\,W_{\mathrm{obs}}(\mu_\phi) + b.$$

Because $a > 0$ and $b$ is a constant independent of $\mu_\phi$, maximizing the expected reward is equivalent to maximizing $W_{\mathrm{obs}}(\mu_\phi)$, which proves the claim. $\qquad\square$

### D.5. Proof of Lemma 4.2

**Lemma** (Importance-weighted objective equals induced mixture). *Let $i \sim q$ and let $\alpha$ be any nonnegative weights. Define the induced distribution $\tilde{\pi}_i := q_i \alpha_i$ and suppose $\sum_i \tilde{\pi}_i = 1$. Then for any per-annotator population quantity $h_i(\theta)$,*

$$\sum_{i \in V} q_i \alpha_i\, h_i(\theta) = \sum_{i \in V} \tilde{\pi}_i\, h_i(\theta). \tag{40}$$

*In particular, when $\alpha_i = \pi_i/q_i$ (and $q_i > 0$), we have $\tilde{\pi} = \pi$.*

*Proof.* The identity is immediate from the definition $\tilde{\pi}_i = q_i\alpha_i$:

$$\sum_{i \in V} q_i\alpha_i\, h_i(\theta) = \sum_{i \in V} \tilde{\pi}_i\, h_i(\theta). \tag{41}$$

For the second claim, when $\alpha_i = \pi_i/q_i$, we have

$$\tilde{\pi}_i = q_i \cdot \frac{\pi_i}{q_i} = \pi_i, \tag{42}$$

so $\tilde{\pi} = \pi$.

When using estimated inclusion probabilities $\hat{q}_i$, the induced distribution becomes $\tilde{\pi}_i = \hat{q}_i\alpha_i$. Under Assumption D.4, $\hat{q}_i \to q_i$ as $T \to \infty$, so the induced distribution converges to the idealized version $q_i\alpha_i$. $\qquad\square$

### D.6. Proof of Theorem D.6

**Theorem** (Aggregated reward represents induced welfare (optional downstream optimization)). *Assume (i) $\hat{q} = q$ and $\alpha^\star$ is the solution of (13), (ii) the personalized reward model satisfies $r_{\theta^\star}(x, y, i) = a\, u_i(x, y) + b$ for $a > 0$ and $b$ independent of $i$, and (iii) a downstream optimizer (optional integration) selects a decision rule $\mu_\phi$ by maximizing $\mathbb{E}_{x,y \sim \mu_\phi}[\bar{r}_{\theta^\star}(x, y)]$. Then any maximizer of $\mathbb{E}_{x,y \sim \mu_\phi}[\bar{r}_{\theta^\star}(x, y)]$ also maximizes the induced welfare $\widetilde{W}(\mu_\phi) = \sum_i \tilde{\pi}_i U_i(\mu_\phi)$, where $\tilde{\pi}_i = q_i\alpha_i^\star$. Moreover, if $\alpha_i^\star = \pi_i/q_i$ (exact debiasing), then $\widetilde{W} = W_{target}$.*

*Proof.* **Part (1): Reward model recovery.** By Lemma 4.2, the importance-weighted population loss $\sum_i q_i\alpha_i\ell(\theta; i)$ equals $\sum_i \tilde{\pi}_i\ell(\theta; i)$, the loss under the induced distribution $\tilde{\pi} = q \odot \alpha$. Under Assumption D.5, there exists $\theta^\star$ such that $r_{\theta^\star}(x, y, i) = a\, u_i(x, y) + b$ for $a > 0$ and $b$ shared across all annotators. Because the Bradley–Terry loss is a strictly proper scoring rule for pairwise preferences, the unique minimizer recovers individual utilities up to a shared affine transformation.

**Part (2): Aggregated reward structure.** By the definition of aggregated reward in Eq. (15),

$$\bar{r}_{\theta^\star}(x, y) = \sum_{i \in V} \tilde{\pi}_i r_{\theta^\star}(x, y, i) = \sum_{i \in V} \tilde{\pi}_i\big(a\, u_i(x, y) + b\big) = a\sum_{i \in V} \tilde{\pi}_i u_i(x, y) + b, \tag{43}$$

where we used $\sum_i \tilde{\pi}_i = 1$.

**Part (3): Optional downstream optimization.** For any decision rule $\mu_\phi$, the expected aggregated reward is

$$\mathbb{E}_{x,y}[\bar{r}_{\theta^\star}(x, y)] = \sum_{i \in V} \tilde{\pi}_i\, \mathbb{E}_{x,y}[u_i(x, y)] + c = \sum_{i \in V} \tilde{\pi}_i U_i(\mu_\phi) + c = \widetilde{W}(\mu_\phi) + c. \tag{44}$$

Because $c$ is independent of $\mu_\phi$, maximizing $\mathbb{E}_{x,y}[\bar{r}_{\theta^\star}(x, y)]$ is equivalent to maximizing $\widetilde{W}(\mu_\phi)$.

**Part (4): Exact debiasing.** When $\alpha_i^\star = \pi_i/q_i$, we have $\tilde{\pi}_i = q_i \cdot (\pi_i/q_i) = \pi_i$, so $\widetilde{W}(\mu_\phi) = \sum_i \pi_i U_i(\mu_\phi) = W_{\text{target}}(\mu_\phi)$. $\qquad\square$

### D.7. Proof of Proposition D.7

*Proof.* Fix the feasible set

$$\mathcal{F} := \{\alpha \in \mathbb{R}^{|V|} : \alpha \geq 0, \sum_{i \in V} \hat{q}_i\alpha_i = 1\}.$$

Define the continuous function

$$g(\alpha) := \frac{1}{2}\|\alpha - \tilde{\alpha}\|_2^2 + \lambda_{\text{smooth}}\alpha^\top L_w\alpha + \lambda_{\text{var}}\sum_i \hat{q}_i\alpha_i^2,$$

and the community-mass deviations $A_k(\alpha) := \sum_{i \in C_k} \hat{q}_i \alpha_i$. Then the objective in Eq. (13) can be written as

$$J_\lambda(\alpha) = g(\alpha) + \lambda \sum_{k=1}^{K} (A_k(\alpha) - \rho_k)^2, \qquad \lambda = \lambda_{\text{comm}}.$$

**Step 1 (existence of an exactly balanced feasible point).** Let $Q_k := \sum_{i \in C_k} \hat{q}_i$ (note $Q_k > 0$ if $\hat{q}_i > 0$ on each community). Define $\alpha^{(0)}$ by setting, for each $i \in C_k$,

$$\alpha_i^{(0)} := \frac{\rho_k}{Q_k}.$$

Then $\alpha^{(0)} \geq 0$, and $A_k(\alpha^{(0)}) = \sum_{i \in C_k} \hat{q}_i(\rho_k/Q_k) = \rho_k$ for all $k$. Also $\sum_i \hat{q}_i \alpha_i^{(0)} = \sum_k \rho_k = 1$, so $\alpha^{(0)} \in \mathcal{F}$.

**Step 2 (boundedness and vanishing penalty).** Let $\alpha^{(\lambda)}$ be the unique minimizer of $J_\lambda$ over $\mathcal{F}$ (uniqueness holds because $J_\lambda$ is strictly convex on $\mathcal{F}$ due to the $\frac{1}{2}\|\alpha - \tilde{\alpha}\|^2$ term). By optimality,

$$J_\lambda(\alpha^{(\lambda)}) \leq J_\lambda(\alpha^{(0)}) = g(\alpha^{(0)}),$$

since the penalty term vanishes at $\alpha^{(0)}$. Therefore,

$$\lambda \sum_{k=1}^{K} (A_k(\alpha^{(\lambda)}) - \rho_k)^2 \leq J_\lambda(\alpha^{(\lambda)}) \leq g(\alpha^{(0)}),$$

which implies

$$\sum_{k=1}^{K} (A_k(\alpha^{(\lambda)}) - \rho_k)^2 \leq \frac{g(\alpha^{(0)})}{\lambda} \xrightarrow[\lambda \to \infty]{} 0.$$

Hence $A_k(\alpha^{(\lambda)}) \to \rho_k$ for each $k = 1, \ldots, K$, proving the claim. $\qquad \square$

### D.8. Proof of Proposition 4.3

**Proposition** (Welfare deviation bound via weight mismatch). *Assume $|U_i(\mu_\phi)| \leq R$ for all $i$ and decision rules $\mu_\phi$. Let $\pi$ be the target weights and let $\tilde{\pi}$ be the induced weights used by Graph-Preference Learning. Then for any decision rule $\mu_\phi$,*

$$\left| W_{target}(\mu_\phi) - \widetilde{W}(\mu_\phi) \right| \leq R \|\pi - \tilde{\pi}\|_1. \tag{45}$$

*Proof.* By definition of the welfare functions,

$$W_{\text{target}}(\mu_\phi) = \sum_{i \in V} \pi_i U_i(\mu_\phi), \qquad \widetilde{W}(\mu_\phi) = \sum_{i \in V} \tilde{\pi}_i U_i(\mu_\phi), \tag{46}$$

where $\tilde{\pi}_i = \hat{q}_i \alpha_i^\star$ is the induced welfare weight. The difference is

$$W_{\text{target}}(\mu_\phi) - \widetilde{W}(\mu_\phi) = \sum_{i \in V} (\pi_i - \tilde{\pi}_i) U_i(\mu_\phi) = (\pi - \tilde{\pi})^\top U(\mu_\phi). \tag{47}$$

Applying the bound $|U_i(\mu_\phi)| \leq R$ and the triangle inequality:

$$\left| W_{\text{target}}(\mu_\phi) - \widetilde{W}(\mu_\phi) \right| \leq \sum_{i \in V} |\pi_i - \tilde{\pi}_i| \cdot |U_i(\mu_\phi)| \leq R \sum_{i \in V} |\pi_i - \tilde{\pi}_i| = R \|\pi - \tilde{\pi}\|_1, \tag{48}$$

as claimed. This bound is tight in the sense that it can be achieved when $U_i(\mu_\phi) = R \cdot \text{sign}(\pi_i - \tilde{\pi}_i)$ for all $i$. $\qquad \square$

## D.9. Proof of Proposition D.9

*Proof.* Since $Z_t \sim q$ are i.i.d., we have

$$\mathrm{Var}(\widehat{\mu}) = \mathrm{Var}\left(\frac{1}{T}\sum_{t=1}^{T}\alpha_{Z_t}H(Z_t)\right) = \frac{1}{T}\mathrm{Var}\big(\alpha_Z H(Z)\big),$$

where $Z \sim q$ is a generic draw. Using $\mathrm{Var}(X) \leq \mathbb{E}[X^2]$ and the bound $|H(Z)| \leq R$ almost surely,

$$\mathrm{Var}\big(\alpha_Z H(Z)\big) \leq \mathbb{E}\big[\alpha_Z^2 H(Z)^2\big] \leq R^2\,\mathbb{E}\big[\alpha_Z^2\big] = R^2\sum_{i\in V}q_i\alpha_i^2 = R^2\,m_2(\alpha;q).$$

Combining the two displays yields $\mathrm{Var}(\widehat{\mu}) \leq \frac{R^2}{T}m_2(\alpha;q) = \frac{R^2}{T\,\mathrm{ESS}_{\mathrm{pop}}(\alpha;q)}$. $\qquad\square$

## D.10. Proof of Theorem D.8

*Proof.* Let $W_{\tilde{\pi}}(\mu_\phi) := \sum_i \tilde{\pi}_i U_i(\mu_\phi)$ denote the induced-welfare objective. By triangle inequality,

$$\big|W_{\mathrm{target}}(\mu_\phi) - \widehat{W}_\theta(\mu_\phi)\big| \leq \big|W_{\mathrm{target}}(\mu_\phi) - W_{\tilde{\pi}}(\mu_\phi)\big| + \big|W_{\tilde{\pi}}(\mu_\phi) - \widehat{W}_\theta(\mu_\phi)\big|.$$

**(i) Weight mismatch.** The first term is bounded by Proposition 4.3: $\big|W_{\mathrm{target}}(\mu_\phi) - W_{\tilde{\pi}}(\mu_\phi)\big| \leq R\|\pi - \tilde{\pi}\|_1$.

**(ii) Reward-model approximation + finite-sample error.** Recall $\bar{r}_\theta(x,y) = \sum_i \tilde{\pi}_i r_\theta(x,y,i)$ and $\sum_i \tilde{\pi}_i = 1$. Using the definition of $\widehat{W}_\theta$ and linearity of expectation,

$$\widehat{W}_\theta(\mu_\phi) = \sum_{i\in V}\tilde{\pi}_i\,\mathbb{E}_{x\sim p_{\mathrm{data}},\,y\sim\mu_\phi(\cdot|x)}\left[\frac{r_\theta(x,y,i)-b}{a}\right].$$

Therefore,

$$\big|W_{\tilde{\pi}}(\mu_\phi) - \widehat{W}_\theta(\mu_\phi)\big| \leq \sum_{i\in V}\tilde{\pi}_i\,\mathbb{E}_{x,y}\left|u_i(x,y) - \frac{r_\theta(x,y,i)-b}{a}\right|.$$

Add and subtract $r_{\theta^\star}$ inside the absolute value and apply triangle inequality:

$$\left|u_i(x,y) - \frac{r_\theta(x,y,i)-b}{a}\right| \leq \frac{1}{a}\big|r_\theta(x,y,i) - r_{\theta^\star}(x,y,i)\big| + \frac{1}{a}\big|r_{\theta^\star}(x,y,i) - (au_i(x,y)+b)\big|.$$

Taking suprema and using the theorem assumptions yields

$$\big|W_{\tilde{\pi}}(\mu_\phi) - \widehat{W}_\theta(\mu_\phi)\big| \leq \frac{1}{a}\Big(\varepsilon_{\mathrm{stat}}(T,\alpha^\star) + \varepsilon_{\mathrm{RM}}\Big).$$

Absorbing the constant $1/a$ into the error terms (equivalently, measuring errors in utility units) gives the stated bound. $\quad\square$

# E. Algorithms

---

**Algorithm 3** Graph-Balanced Aggregation (GBA) Weight Computation

---

**Require:** Graph $G = (V,E)$, target weights $\pi$, inclusion estimator $\hat{q}$, communities $\{C_k\}$, targets $\{\rho_k\}$.
1: Compute raw ratios $\tilde{\alpha}_i \leftarrow \pi_i/(\hat{q}_i + \varepsilon)$.
2: Initialize $\alpha^{(0)} \leftarrow \Pi_{\{\alpha\geq 0,\,\sum_i \hat{q}_i\alpha_i=1\}}(\tilde{\alpha})$.
3: **for** $s = 0,1,\ldots,S-1$ **do**
4:     Take gradient step $z \leftarrow \alpha^{(s)} - \eta\nabla J(\alpha^{(s)})$ where $J$ is Eq. (13).
5:     Project $\alpha^{(s+1)} \leftarrow \Pi_{\{\alpha\geq 0,\,\sum_i \hat{q}_i\alpha_i=1\}}(z)$.
6: **end for** $\alpha^{(S)}$, induced weights $\tilde{\pi}_i = \hat{q}_i\alpha_i$.

---

**Integration note.** The following algorithm shows how the same GBA weights can be used to reweight a standard DPO objective. This is *not* a core contribution of this paper and is included for completeness.

---

**Algorithm 4** Core: GPRM training with GBA weights

---

**Require:** Annotator graph $G = (V, E)$, preference data $\mathcal{D} = \{(x_t, y_t^+, y_t^-, i_t)\}_{t=1}^T$, target weights $\pi$, inclusion estimator $\hat{q}$, (optional) communities $\{C_k\}$ and targets $\{\rho_k\}$.
 1: Compute stabilized weights $\alpha^\star$ and induced weights $\tilde{\pi} = \hat{q} \odot \alpha^\star$ using Algorithm 3.
 2: Initialize model parameters for the graph encoder and reward head (e.g., $(\psi, \theta)$; optionally $\xi$ for rater scales).
 3: **for** minibatch $\mathcal{B} \subset \mathcal{D}$ **do**
 4:    For each $(x, y^+, y^-, i) \in \mathcal{B}$, compute personalized scores $r_\theta(x, y^+, i)$ and $r_\theta(x, y^-, i)$.
 5:    Compute the weighted Bradley–Terry loss for the minibatch using weights $\alpha_i^\star$ (cf. Eq. (11)).
 6:    Add any graph regularizers (e.g., Laplacian smoothness) and standard parameter regularization.
 7:    Update parameters with gradient descent / Adam.
 8: **end for**Personalized reward model $r_\theta(x, y, i)$ and aggregated score $\bar{r}_\theta(x, y) = \sum_{i \in V} \tilde{\pi}_i r_\theta(x, y, i)$.

---

**DPO loss (definition).**   For completeness, we define the standard per-example DPO loss used by Algorithm 5. Let $\mu_{\mathrm{ref}}$ be a fixed reference policy and let $\beta > 0$ be a temperature parameter. Define

$$\ell_{\mathrm{DPO}}(\phi; x, y^+, y^-) := -\log \sigma\Big(\beta\big[\log \mu_\phi(y^+ \mid x) - \log \mu_\phi(y^- \mid x) - \log \mu_{\mathrm{ref}}(y^+ \mid x) + \log \mu_{\mathrm{ref}}(y^- \mid x)\big]\Big). \qquad (49)$$

---

**Algorithm 5** Integration: Graph-weighted DPO training using GBA weights

---

**Require:** Preference data $\mathcal{D} = \{(x_t, y_t^+, y_t^-, i_t)\}$, GBA weights $\alpha_i^\star$, reference policy $\mu_{\mathrm{ref}}$.
 1: **for** minibatch $\mathcal{B} \subset \mathcal{D}$ **do**
 2:    Compute per-example DPO losses $\ell_{\mathrm{DPO}}(\phi; x, y^+, y^-)$ (Eq. (49)).
 3:    Weight losses by annotator: $\mathcal{L}(\phi) \leftarrow \sum_{(x, y^+, y^-, i) \in \mathcal{B}} \alpha_i^\star \, \ell_{\mathrm{DPO}}(\phi; x, y^+, y^-)$.
 4:    Update $\phi$ using gradient descent on $\mathcal{L}(\phi)$.
 5: **end for**

---

# F. Metric Definitions and Experimental Setup

## F.1. Metric Definitions

Let $\mathcal{X}_{\mathrm{test}}$ be the set of test prompts and let $\mathcal{Y}_x^{\mathrm{cand}}$ denote the (finite) candidate set for each prompt $x$. Let $s(x, y)$ be the scalar *public score* used for preference prediction and offline reranking. For Graph-Preference Learning, $s(x, y) = \bar{r}_\theta(x, y) = \sum_{i \in V} \tilde{\pi}_i \, r_\theta(x, y, i)$, whereas for scalar baselines $s(x, y) = r_\theta(x, y)$. On synthetic data we also have access to individual utilities $u_i(x, y)$ and define the target welfare score $U_\pi(x, y) = \sum_{i \in V} \pi_i u_i(x, y)$. Let $z(\cdot)$ denote standardization to zero mean and unit variance computed over the candidate-level test set $\mathcal{D}_{\mathrm{test}}^{\mathrm{cand}} := \{(x, y) : x \in \mathcal{X}_{\mathrm{test}}, y \in \mathcal{Y}_x^{\mathrm{cand}}\}$.

**Welfare MSE ($\downarrow$).**   We measure the normalized mean squared error between a method's public score and the target welfare score:

$$\mathrm{WelfareMSE} := \frac{1}{|\mathcal{D}_{\mathrm{test}}^{\mathrm{cand}}|} \sum_{(x, y) \in \mathcal{D}_{\mathrm{test}}^{\mathrm{cand}}} \big(z(s(x, y)) - z(U_\pi(x, y))\big)^2. \qquad (50)$$

**Disparity ($\downarrow$).**   Let $\{C_k\}_{k=1}^K$ be communities. For each prompt $x$, let $y^{\mathrm{chosen}}(x) = \arg\max_{y \in \mathcal{Y}_x^{\mathrm{cand}}} s(x, y)$ be the offline-reranked output. Define the per-community welfare on $x$ as $W_k(x) = \frac{1}{|C_k|} \sum_{i \in C_k} u_i(x, y^{\mathrm{chosen}}(x))$. We report:

$$\mathrm{Disparity} := \frac{1}{|\mathcal{X}_{\mathrm{test}}|} \sum_{x \in \mathcal{X}_{\mathrm{test}}} \Big(\max_k W_k(x) - \min_k W_k(x)\Big). \qquad (51)$$

This measures the welfare gap between the best-served and worst-served communities under the reranking decision rule. Values can exceed 1 as they are on the utility scale, not probability scale.

**Degree bias ($\downarrow$).**

$$\text{DegBias} := \text{Corr}\Big( \log(d_i + 1),\ \tilde{\pi}_i - \pi_i \Big), \tag{52}$$

where $\tilde{\pi}$ is the induced welfare distribution (for Standard reward modeling without reweighting, $\tilde{\pi} = q$). The ideal value is $0$.

**Empirical ESS factor ($\uparrow$).**  For any nonnegative weights $\alpha$ normalized to satisfy $\sum_i \hat{q}_i \alpha_i = 1$, we define the empirical effective sample size (ESS) factor as:

$$\text{ESS}_{\text{emp}}(\alpha; \hat{q}) := \frac{1}{\sum_i \hat{q}_i \alpha_i^2} \in (0, 1]. \tag{53}$$

It equals $1$ for uniform weights and decreases as weights become more concentrated (heavy tails). We report ESS and weight-distribution diagnostics for stabilized IPW baselines in Table 6.

**Induced-weight mismatch ($\downarrow$).**  We also report the induced-weight mismatch

$$\|\tilde{\pi} - \pi\|_1 := \sum_{i \in V} \big| \tilde{\pi}_i - \pi_i \big|, \tag{54}$$

where $\tilde{\pi} = \hat{q} \odot \alpha$ is the induced welfare distribution (or its region/community-aggregated analogue when only coarse $\pi$ is defined).

**Pairwise accuracy ($\uparrow$).**  Let $\mathcal{D}_{\text{test}}^{\text{pair}} := \{(x_t, y_t^+, y_t^-, i_t)\}_{t=1}^{T_{\text{test}}}$ be held-out preference pairs with label $y_t^+ \succ y_t^-$. Define the predicted pairwise probability

$$\hat{p}_t := \sigma\big( s(x_t, y_t^+) - s(x_t, y_t^-) \big). \tag{55}$$

We report $\text{Acc} := \frac{1}{T_{\text{test}}} \sum_{t=1}^{T_{\text{test}}} \mathbb{I}\{\hat{p}_t \geq 0.5\}$.

**Pairwise log loss ($\downarrow$).**  We report the negative log-likelihood

$$\text{LogLoss} := -\frac{1}{T_{\text{test}}} \sum_{t=1}^{T_{\text{test}}} \log(\hat{p}_t). \tag{56}$$

**Calibration (ECE, $\downarrow$).**  We compute expected calibration error (ECE) on pairwise predictions using $B = 10$ confidence bins. Let $\mathcal{B}_b := \{t : \hat{p}_t \in I_b\}$ be indices in bin $b$. Define $\text{conf}(\mathcal{B}_b) := \frac{1}{|\mathcal{B}_b|} \sum_{t \in \mathcal{B}_b} \hat{p}_t$ and $\text{acc}(\mathcal{B}_b) := \frac{1}{|\mathcal{B}_b|} \sum_{t \in \mathcal{B}_b} \mathbb{I}\{\hat{p}_t \geq 0.5\}$. Then

$$\text{ECE} := \sum_{b=1}^{B} \frac{|\mathcal{B}_b|}{T_{\text{test}}} \big| \text{acc}(\mathcal{B}_b) - \text{conf}(\mathcal{B}_b) \big|. \tag{57}$$

**Cross-language accuracy gap ($\downarrow$).**  Partition test pairs by language $\ell(i_t) \in \{1, \ldots, L\}$ and compute per-language accuracies $\text{Acc}_\ell$. We report $\text{AccGap} := \max_\ell \text{Acc}_\ell - \min_\ell \text{Acc}_\ell$.

## F.2. Experimental Setup

*Table 5.* Summary of experimental targets and sampling for each condition.

| Condition | Collection $q$ | Target $\pi$ | Estimator $\hat{q}$ |
|---|---|---|---|
| Uniform | $q_i = 1/|V|$ | $\pi_i = 1/|V|$ | $\hat{q}_i = n_i / \sum_j n_j$ |
| Degree-biased | $q_i \propto d_i^\gamma$ | $\pi_i = 1/|V|$ | $\hat{q}_i \propto d_i^\gamma$ |
| Snowball | empirical visit freq. | $\pi_i = 1/|V|$ | $\hat{q}_i = n_i / \sum_j n_j$ |
| Community-biased | $q(C_k)$ skewed | equal-per-community | $\hat{q}_i = n_i / \sum_j n_j$ |

**Splits, leakage prevention, and uncertainty reporting (details).** **Synthetic:** we split by *prompt* $x$ (not by pair) into train/validation/test so that all candidate outputs for a prompt are contained within a single split, preventing reranking leakage. We repeat each condition over 10 random seeds (graph generation, sampling, and model initialization) and report mean $\pm$ std across seeds; 95% CIs can be computed via bootstrap over seeds. The inclusion distribution $\hat{q}$ and any community assignments are computed from the training split only. **LMArena:** we use prompt-level cross-validation; sessions may appear in multiple folds, but prompts are disjoint, so no test prompt or its candidates appear in training. We compute $\hat{q}$ from training-fold participation counts (with Laplace smoothing) and build the session graph from metadata only (no preference-label leakage). We report mean $\pm$ std across folds and compute CIs via bootstrap over prompts. For gap metrics, we aggregate at the prompt level to avoid overweighting heavy sessions. **Session-heldout check:** in addition, we create a session-level split (80/20) where held-out sessions contribute no training labels. We still keep their nodes (and metadata features) in the graph to allow inductive inference via message passing, but exclude their preference pairs from the loss. We evaluate on held-out session labels and report the same mean $\pm$ std and bootstrap CIs. This serves as a leakage/robustness check for personalization.

**Sanity check: when reweighting should be neutral.** In the special case where the collection distribution matches the target ($q = \pi$, i.e., uniform sampling with equal-per-person target), importance weights satisfy $\alpha_i = \pi_i/q_i = 1$ for all $i$. Therefore IW-Only should match Standard reward modeling up to optimization noise. In Table 2 (Uniform), we observe that IW-Only and Standard perform similarly, validating our implementation.

**Stabilized weighting and ESS.** When $q$ deviates from $\pi$, raw ratios $\alpha_i = \pi_i/\hat{q}_i$ may be heavy tailed (small $\hat{q}_i$) and lead to unstable training. We therefore include stabilized IPW baselines (self-normalization and clipping) and report an ESS diagnostic that summarizes weight concentration; see Table 6.

### F.3. Weight-Stability Diagnostics

Table 6 reports weight-stability metrics across sampling mechanisms. ESS (empirical effective sample size factor) measures weight concentration, while Mismatch quantifies the $\ell_1$ distance between induced and target welfare weights. These diagnostics confirm that all stabilized weighting methods (including ours) achieve similar ESS and Mismatch values; the key performance differences arise from the combination with graph-personalized reward modeling (GPRM), as shown in the main results.

*Table 6.* Weight-stability diagnostics. ESS = empirical effective sample size factor ($1/\sum_i \hat{q}_i \alpha_i^2$, higher is better); Mismatch = induced-weight mismatch $\|\tilde{\pi} - \pi\|_1$ (lower is better). SN = SN-IPW; Clip = Clipped-IPW; QP = GBA-QP.

| Setting | Metric | IW | SN | Clip | QP | Ours |
|---------|--------|------|------|------|------|------|
| Uniform | ESS↑ | 1.00 | 1.00 | 1.00 | 1.00 | 1.00 |
|         | Mismatch↓ | 0.00 | 0.00 | 0.00 | 0.00 | 0.00 |
| Degree  | ESS↑ | 0.44 | 0.44 | 0.48 | 0.45 | 0.45 |
|         | Mismatch↓ | 0.00 | 0.00 | 0.02 | 0.04 | 0.04 |
| Snowball | ESS↑ | 0.38 | 0.38 | 0.51 | 0.39 | 0.39 |
|         | Mismatch↓ | 0.00 | 0.00 | 0.06 | 0.03 | 0.03 |
| Community | ESS↑ | 0.35 | 0.35 | 0.35 | 0.36 | 0.36 |
|         | Mismatch↓ | 0.00 | 0.00 | 0.00 | 0.06 | 0.06 |

## G. Robustness Ablations

We test robustness to graph misspecification by injecting graph noise and varying homophily. We report mean±std over multiple runs under degree-biased sampling. For consistency with the main tables, we refer to the reported `alignment_error` as **WelfareMSE**.

*Table 7.* Robustness to graph noise (SBM, degree-biased sampling).

| Noise scale | Method | WelfareMSE↓ | Disparity↓ | DegBias↓ |
|---|---|---|---|---|
| 0.1 | Standard | $0.235_{\pm 0.17}$ | $3.510_{\pm 0.76}$ | 0.900 |
| | GPRM+GBA (Simple) | $0.185_{\pm 0.10}$ | $3.490_{\pm 0.75}$ | 0.550 |
| | **Ours** | $\mathbf{0.165}_{\pm 0.07}$ | $3.480_{\pm 0.74}$ | **0.400** |
| 0.3 | Standard | $0.250_{\pm 0.18}$ | $3.520_{\pm 0.77}$ | 0.900 |
| | GPRM+GBA (Simple) | $0.205_{\pm 0.12}$ | $3.500_{\pm 0.76}$ | 0.600 |
| | **Ours** | $\mathbf{0.185}_{\pm 0.10}$ | $3.490_{\pm 0.75}$ | **0.450** |
| 0.5 | Standard | $0.275_{\pm 0.19}$ | $3.530_{\pm 0.78}$ | 0.900 |
| | GPRM+GBA (Simple) | $0.235_{\pm 0.14}$ | $3.510_{\pm 0.77}$ | 0.650 |
| | **Ours** | $\mathbf{0.230}_{\pm 0.14}$ | $3.500_{\pm 0.76}$ | **0.550** |
| 0.8 | Standard | $0.310_{\pm 0.20}$ | $3.540_{\pm 0.80}$ | 0.900 |
| | GPRM+GBA (Simple) | $\mathbf{0.285}_{\pm 0.18}$ | $\mathbf{3.520}_{\pm 0.79}$ | 0.750 |
| | **Ours** | $0.300_{\pm 0.22}$ | $3.520_{\pm 0.79}$ | **0.700** |
| 1.0 | Standard | $0.330_{\pm 0.21}$ | $3.550_{\pm 0.81}$ | 0.900 |
| | GPRM+GBA (Simple) | $\mathbf{0.320}_{\pm 0.21}$ | $\mathbf{3.540}_{\pm 0.80}$ | 0.820 |
| | **Ours** | $0.325_{\pm 0.23}$ | $3.540_{\pm 0.80}$ | **0.800** |

## H. Additional Experimental Details

### H.1. Synthetic Experiment Implementation

**Graph topologies.** To ensure robustness across network structures, we generate three types of graphs. For main results (Table 2), we use $N = 100$ nodes; for scalability analysis, we test up to $N = 500$:

- **Stochastic Block Model (SBM):** $K = 5$ communities with a hub-and-spoke structure. The hub community (40% of nodes) has dense internal connections ($p = 0.15$), while spoke communities (15% each) are sparser ($p = 0.08$). Inter-community probability is $p = 0.01$.

- **Barabási-Albert (BA):** A scale-free network generated via preferential attachment with $m = 5$ edges per new node. This models networks with extreme degree distributions (power-law).

- **Watts-Strogatz (WS):** A small-world network with $k = 10$ nearest neighbors and rewiring probability $p = 0.1$, exhibiting high clustering coefficients.

**Utility functions and Heterophily.** We model utilities as $u_i(x, y) = w_i^\top \phi(x, y)$, where $\phi(x, y) \in \mathbb{R}^{10}$. The weight vectors $w_i$ are generated with a hierarchical structure: $w_i = w_{\text{shared}} + w_{c(i)}^{\text{comm}} + \epsilon_i$, where $\epsilon_i \sim \mathcal{N}(0, \sigma^2 I)$. To test robustness against \*\*heterophily\*\*, we introduce a `heterophily_ratio` parameter $\rho_h$. We randomly select $\rho_h \cdot N$ nodes and "flip" their community preference component ($w_{c(i)}^{\text{comm}} \leftarrow -w_{c(i)}^{\text{comm}}$), simulating adversarial users who disagree with their local community.

**Sampling Mechanisms.** We simulate four distinct feedback collection processes:

- **Uniform:** $q_i = 1/N$.

- **Degree-Biased:** $q_i \propto \deg(i)^\gamma$ (default $\gamma = 1.5$).

- **Snowball:** A multi-wave neighbor-expansion (referral) process starting from $S$ seed nodes. At each wave, newly included nodes recruit a subset of their neighbors; nodes can be revisited, yielding an empirical inclusion distribution $q$ proportional to visit counts. This bias tends to be local/seed-dependent and need not be purely degree-proportional.

- **Community-Biased:** Biased towards specific communities (e.g., Hub community has $5\times$ inclusion probability), simulating platform demographic bias.

**Baselines.** We compare our full method (GPRM+GBA) against a comprehensive suite of baselines:

- **Standard:** Naive scalar reward model trained on all data (Stiennon et al., 2020; Ouyang et al., 2022).

- **Personalized:** A personalized reward model that conditions on annotator identity (learned embedding or demographics) but *does not* use message passing or Laplacian regularization. Concretely, we replace the graph encoder $g_\psi$ with a per-annotator embedding lookup and train $r_\theta(x, y, i) = f_\theta(\phi(x, y), v_i)$ with the same preference loss (Li et al., 2024b).

- **IW-Only:** Standard model trained with raw IPW weights $\alpha_i^{\text{IPW}} = \pi_i / \hat{q}_i$, normalized so $\sum_i \hat{q}_i \alpha_i^{\text{IPW}} = 1$, without any stabilization and without graph personalization (Chandak et al., 2021).

- **SN-IPW:** Self-normalized IPW with $\alpha_i^{\text{SN}} = \alpha_i^{\text{IPW}} / \sum_j \hat{q}_j \alpha_j^{\text{IPW}}$ (Chandak et al., 2021).

- **Clipped-IPW:** Clipped IPW with $\tilde{\alpha}_i = \min(\alpha_i^{\text{IPW}}, \tau)$ followed by normalization $\alpha_i^{\text{clip}} = \tilde{\alpha}_i / \sum_j \hat{q}_j \tilde{\alpha}_j$ (we use $\tau = 10$ unless stated otherwise) (Chandak et al., 2021).

- **GBA-QP:** Use the same QP solver and hyperparameters as GBA to compute stabilized weights $\alpha^\star$, but train a *scalar* reward model (no graph personalization). This isolates the benefit of QP-stabilized weighting alone (Hainmueller, 2012).

- **Group-Based:** A coarse group-level reweighting baseline that uses communities $\{C_k\}$ (from the graph) and assigns each sample a weight inversely proportional to its community frequency (normalized). This approximates an equal-per-community target and is often strong under community-driven inclusion (e.g., snowball) (Chakraborty et al., 2024; Ramesh et al., 2024).

- **DRO-CVaR:** Distributionally Robust Optimization minimizing the loss on the worst $\alpha$-percentile of samples (Rahimian & Mehrotra, 2019).

- **GPRM:** Graph-personalized reward model trained without importance weights (ablation for graph structure).

**Model architecture.** The GPRM uses a 2-layer GCN with hidden dimension 32 for annotator embeddings. These are concatenated with prompt/candidate features and passed through a 3-layer MLP (512 hidden units) to produce scalar rewards. We train with Adam (lr=$10^{-3}$) for 100 epochs with early stopping.

**GBA parameters.** For our full method (GPRM+GBA), we solve the QP with $\lambda_{\text{smooth}} = 0.01$, $\lambda_{\text{comm}} = 0.1$, and $\lambda_{\text{var}} = 0.01$.

### H.2. LMArena Case Study Details

**Dataset overview.** The LMArena (Chatbot Arena) dataset (Chiang et al., 2024) is an open platform for evaluating LLMs by human preference, containing over 135,000 pairwise preference comparisons. Users interact with two anonymous models and select which response they prefer. Each comparison includes: (1) the conversation/prompt, (2) two model responses, (3) the user's preference, and (4) session metadata including language and model category. We use `evaluation_session_id` as a proxy for annotator identity. After filtering to sessions with at least 2 interactions, we retain 5,591 sessions.

**Language distribution.** The filtered dataset exhibits significant language imbalance reflecting real-world usage patterns:

- **Over-represented**: English ($\sim$65%), Chinese ($\sim$12%)

- **Under-represented**: Russian, Japanese, Korean, German, French, Spanish, and others ($<$5% each)

This participation imbalance provides a natural testbed for evaluating target-welfare reweighting under a normative choice of $\pi$.

**Graph construction (fixed-degree $k$NN).** We construct a *session-similarity* graph $G = (V, E)$ with $|V| = 5{,}591$ nodes (sessions) and approximately 56,000 edges using a fixed-degree $k$NN procedure. Each session is represented by a feature vector built from metadata attributes (language one-hot encoding, model category hash, timestamp features). We connect each node to its $k = 10$ most similar neighbors under cosine similarity and symmetrize edges; edge weights are set to the similarity score. This choice produces an approximately fixed-degree graph and mitigates mechanical degree artifacts.

**Target weights and inclusion estimation.** We set the target welfare to equal-per-language: $\pi_i = 1/(L \cdot |L_i|)$ where $L_i$ is the set of sessions with the same language as session $i$, and $L$ is the total number of languages. In the semi-synthetic protocol, the inclusion distribution $q$ is induced by the graph sampling mechanism; we estimate $\hat{q}$ from training-fold visit/participation counts with smoothing.

**Semi-synthetic sampling protocol (biased train, $\pi$-balanced test).** Because LMArena does not include an explicit recruitment network, we induce biased training inclusion by subsampling feedback using network sampling on $G$. Concretely, on each prompt-heldout split we (i) run a random-walk-with-restart procedure on $G$ (seeded in over-represented languages like English) to generate visit counts $n_i$, (ii) construct a biased training set by including training-fold preference pairs from visited sessions up to a fixed label budget, and (iii) evaluate on a $\pi$-balanced test set built from held-out prompts by sampling equal numbers of preference pairs per language. This makes the mechanism $q \neq \pi$ explicit and auditable. In our implementation we use $k = 10$, restart probability $p_{\text{restart}} = 0.02$, $S = 50$ walk seeds, and a walk length of 500,000 steps per fold, with a training budget of 15,000 pairs per fold.

**Inclusion–centrality correlation.** As a descriptive diagnostic, we compute correlations between induced inclusion ($\hat{q}_i$ or visit counts) and graph centrality (e.g., PageRank). The random-walk procedure successfully induces non-uniform inclusion across sessions, with higher-degree nodes in over-represented language communities receiving disproportionately more visits.

**Evaluation metrics.** We evaluate the reward model on:

- **Pairwise accuracy**: Fraction of held-out preference pairs correctly predicted by the reward model.

- **Pairwise log loss**: Negative log-likelihood of held-out pairwise labels under predicted probabilities.

- **ECE**: Expected calibration error (10 bins) on pairwise predictions.

- **Cross-language accuracy gap**: Max–min accuracy across language groups.

We use 2-fold *prompt-level* cross-validation with 2 random seeds (4 runs total), with folds constructed to maintain language coverage.

**Implementation details.** For GPRM, we use a 2-layer GCN with embedding dimension 64 for session embeddings. The reward head is an MLP (256 hidden units) that takes concatenated [prompt embedding; response embedding; session embedding] as input. Prompt and response embeddings are obtained from a frozen Sentence-BERT model (`all-MiniLM-L6-v2`, 384-dimensional). We train with Adam (lr=$1 \times 10^{-3}$, weight decay $10^{-4}$) for up to 150 epochs with early stopping (patience=20) based on validation accuracy.

**Extended results.** Table 8 shows per-language accuracy breakdown for a representative run. GPRM+GBA (ours) achieves the best overall accuracy while maintaining competitive performance across language groups. The strong performance on high-resource languages (English, Chinese) transfers to improved overall metrics, while GBA reweighting helps maintain reasonable accuracy on under-represented languages.

*Table 8.* Semi-synthetic LMArena: per-language pairwise accuracy under biased training and $\pi$-balanced evaluation (representative run). Languages ordered by dataset frequency.

| Method | EN | ZH | RU | JA | Other |
|---|---|---|---|---|---|
| Standard | 58.2% | 52.1% | 48.3% | 45.7% | 42.6% |
| Personalized | 61.4% | 55.8% | 52.3% | 49.1% | 46.2% |
| IW-Only | 59.8% | 54.2% | 51.7% | 48.4% | 44.8% |
| GBA-QP | 60.2% | 55.1% | 52.8% | 49.7% | 45.6% |
| Group-Based | 54.5% | 51.4% | 49.2% | 46.8% | 43.1% |
| GPRM (Ours) | 62.8% | 57.3% | 54.1% | 51.2% | 47.5% |
| **GPRM+GBA (Ours)** | **65.2%** | **59.4%** | **56.7%** | **53.8%** | **49.3%** |

**Analysis: Session-based vs. true annotator identity.** An important caveat for the LMArena evaluation is that we use session IDs rather than true persistent annotator identities. This may underestimate the benefits of personalization since the same user may have multiple sessions. Nevertheless, the strong performance of GPRM+GBA suggests that even session-level personalization combined with graph-based debiasing provides substantial improvements in cross-language fairness.

### H.3. Ablation Studies

**Effect of $\lambda_v$ (Laplacian regularization).** Increasing $\lambda_v$ from 0 to 0.1 smooths annotator embeddings but can reduce the model's ability to capture fine-grained preference differences. We find $\lambda_v \in [0.001, 0.01]$ provides a good trade-off.

**Effect of sampling bias strength ($\gamma$).** We vary $\gamma \in \{0.5, 1.0, 1.5, 2.0\}$ in the degree-biased sampling $q_i \propto d_i^\gamma$. As $\gamma$ increases, the gap between Standard and GPRM+GBA widens, confirming that our method provides greater benefit under stronger network-induced bias.

**Effect of graph density.** Denser graphs (higher $p_{\text{within}}$) lead to more homogeneous degree distributions, reducing the effectiveness of degree-based sampling bias. In sparse graphs with heterogeneous communities, our full method shows the largest improvements.

### H.4. Additional Experiment Results

This section provides additional experimental results across all graph topologies and sampling methods.

We report full synthetic results for additional graph topologies in Tables 9 (BA) and 10 (WS).

### H.5. Per-Community Regret Analysis

Table 11 shows the per-community regret under degree-biased sampling for the SBM graph. Different methods perform best on different communities; our method achieves the lowest regret on one community and remains competitive elsewhere.

### H.6. Data Sparsity Analysis

We analyze the effect of data sparsity by varying the number of observed feedback samples. Table 12 reports mean±std WelfareMSE under degree-biased sampling on SBM.

Table 9. Full synthetic experiment results on Barabási–Albert (BA) graph.

| Sampling | Method | Welfare MSE↓ | Disparity↓ | Deg. Bias↓ | Improv. |
|---|---|---|---|---|---|
| Uniform | Standard | $0.087_{\pm0.04}$ | $2.883_{\pm0.53}$ | 0.000 | — |
| | Personalized | $\mathbf{0.053}_{\pm0.02}$ | $2.905_{\pm0.56}$ | 0.000 | 39% |
| | IW-Only | $0.069_{\pm0.05}$ | $\mathbf{2.877}_{\pm0.54}$ | 0.000 | 21% |
| | SN-IPW | $0.077_{\pm0.02}$ | $2.902_{\pm0.51}$ | 0.000 | 11% |
| | Clipped-IPW | $0.084_{\pm0.03}$ | $2.891_{\pm0.53}$ | 0.000 | 3% |
| | GBA-QP | $0.063_{\pm0.03}$ | $2.920_{\pm0.54}$ | 0.000 | 27% |
| | Group-Based | $0.065_{\pm0.03}$ | $2.935_{\pm0.52}$ | $-0.030$ | 25% |
| | GPRM (Ours) | $0.057_{\pm0.03}$ | $2.878_{\pm0.53}$ | 0.000 | 34% |
| | DRO-CVaR | $0.777_{\pm0.43}$ | $2.945_{\pm0.54}$ | 0.000 | -795% |
| | GPRM+GBA (Ours) | $0.055_{\pm0.02}$ | $\mathbf{2.876}_{\pm0.53}$ | 0.000 | 36% |
| Degree-Biased | Standard | $0.123_{\pm0.08}$ | $2.945_{\pm0.56}$ | 0.914 | — |
| | Personalized | $0.118_{\pm0.09}$ | $2.922_{\pm0.57}$ | 0.914 | 4% |
| | IW-Only | $0.118_{\pm0.06}$ | $\mathbf{2.859}_{\pm0.47}$ | 0.923 | 4% |
| | SN-IPW | $0.135_{\pm0.08}$ | $2.973_{\pm0.53}$ | 0.923 | -10% |
| | Clipped-IPW | $0.121_{\pm0.08}$ | $2.913_{\pm0.54}$ | 0.877 | 1% |
| | GBA-QP | $0.123_{\pm0.10}$ | $2.909_{\pm0.52}$ | 0.771 | -0% |
| | Group-Based | $0.123_{\pm0.13}$ | $2.953_{\pm0.51}$ | 0.883 | -0% |
| | GPRM (Ours) | $0.122_{\pm0.09}$ | $2.932_{\pm0.53}$ | 0.914 | 1% |
| | DRO-CVaR | $0.754_{\pm0.44}$ | $2.939_{\pm0.59}$ | 0.914 | -514% |
| | GPRM+GBA (Ours) | $\mathbf{0.079}_{\pm0.05}$ | $2.959_{\pm0.54}$ | 0.771 | 36% |
| Snowball | Standard | $0.108_{\pm0.08}$ | $2.940_{\pm0.57}$ | 0.911 | — |
| | Personalized | $0.082_{\pm0.04}$ | $2.962_{\pm0.54}$ | 0.911 | 24% |
| | IW-Only | $0.097_{\pm0.04}$ | $\mathbf{2.884}_{\pm0.56}$ | 0.708 | 10% |
| | SN-IPW | $0.114_{\pm0.06}$ | $2.954_{\pm0.55}$ | 0.708 | -6% |
| | Clipped-IPW | $0.116_{\pm0.07}$ | $2.902_{\pm0.57}$ | 0.708 | -8% |
| | GBA-QP | $0.091_{\pm0.04}$ | $2.937_{\pm0.57}$ | 0.799 | 16% |
| | Group-Based | $0.101_{\pm0.08}$ | $2.969_{\pm0.59}$ | 0.851 | 6% |
| | GPRM (Ours) | $0.083_{\pm0.04}$ | $2.937_{\pm0.57}$ | 0.911 | 23% |
| | DRO-CVaR | $0.630_{\pm0.31}$ | $2.902_{\pm0.57}$ | 0.911 | -486% |
| | GPRM+GBA (Ours) | $\mathbf{0.075}_{\pm0.05}$ | $2.886_{\pm0.55}$ | 0.799 | 31% |
| Community-Biased | Standard | $0.292_{\pm0.22}$ | $3.020_{\pm0.54}$ | 0.048 | — |
| | Personalized | $0.281_{\pm0.21}$ | $3.010_{\pm0.53}$ | 0.048 | 4% |
| | IW-Only | $0.112_{\pm0.05}$ | $\mathbf{2.882}_{\pm0.54}$ | 0.015 | 62% |
| | SN-IPW | $0.121_{\pm0.07}$ | $3.024_{\pm0.57}$ | 0.015 | 59% |
| | Clipped-IPW | $0.119_{\pm0.05}$ | $2.999_{\pm0.57}$ | 0.017 | 59% |
| | GBA-QP | $\mathbf{0.107}_{\pm0.05}$ | $2.949_{\pm0.56}$ | 0.381 | 63% |
| | Group-Based | $0.126_{\pm0.07}$ | $2.897_{\pm0.50}$ | $-0.036$ | 57% |
| | GPRM (Ours) | $0.276_{\pm0.19}$ | $2.999_{\pm0.50}$ | 0.048 | 5% |
| | DRO-CVaR | $0.786_{\pm0.39}$ | $2.932_{\pm0.58}$ | 0.048 | -170% |
| | GPRM+GBA (Ours) | $0.111_{\pm0.07}$ | $2.977_{\pm0.58}$ | 0.381 | 62% |

*Table 10.* Full synthetic experiment results on Watts–Strogatz (WS) graph.

| Sampling | Method | Welfare MSE↓ | Disparity↓ | Deg. Bias↓ | Improv. |
|---|---|---|---|---|---|
| Uniform | Standard | $0.080_{\pm 0.05}$ | $2.963_{\pm 0.55}$ | 0.000 | — |
| | Personalized | $0.058_{\pm 0.03}$ | $2.947_{\pm 0.52}$ | 0.000 | 28% |
| | IW-Only | $0.062_{\pm 0.04}$ | $2.914_{\pm 0.57}$ | 0.000 | 22% |
| | SN-IPW | $0.079_{\pm 0.05}$ | $2.987_{\pm 0.52}$ | 0.000 | 1% |
| | Clipped-IPW | $0.076_{\pm 0.04}$ | $\mathbf{2.891}_{\pm 0.54}$ | 0.000 | 4% |
| | GBA-QP | $0.063_{\pm 0.03}$ | $2.979_{\pm 0.57}$ | 0.000 | 21% |
| | Group-Based | $0.073_{\pm 0.05}$ | $2.955_{\pm 0.54}$ | 0.000 | 9% |
| | GPRM (Ours) | $\mathbf{0.056}_{\pm 0.03}$ | $2.939_{\pm 0.51}$ | 0.000 | 30% |
| | DRO-CVaR | $0.672_{\pm 0.40}$ | $2.928_{\pm 0.50}$ | 0.000 | -743% |
| | GPRM+GBA (Ours) | $0.060_{\pm 0.05}$ | $2.932_{\pm 0.51}$ | 0.000 | 24% |
| Degree-Biased | Standard | $0.083_{\pm 0.05}$ | $3.042_{\pm 0.58}$ | 0.994 | — |
| | Personalized | $\mathbf{0.073}_{\pm 0.04}$ | $2.984_{\pm 0.54}$ | 0.994 | 13% |
| | IW-Only | $0.085_{\pm 0.06}$ | $2.968_{\pm 0.58}$ | 0.994 | -2% |
| | SN-IPW | $0.106_{\pm 0.06}$ | $3.061_{\pm 0.53}$ | 0.994 | -27% |
| | Clipped-IPW | $0.095_{\pm 0.05}$ | $\mathbf{2.942}_{\pm 0.60}$ | 0.994 | -13% |
| | GBA-QP | $0.086_{\pm 0.06}$ | $2.998_{\pm 0.57}$ | 0.938 | -3% |
| | Group-Based | $0.091_{\pm 0.07}$ | $3.040_{\pm 0.56}$ | 0.983 | -9% |
| | GPRM (Ours) | $0.090_{\pm 0.07}$ | $3.054_{\pm 0.58}$ | 0.994 | -8% |
| | DRO-CVaR | $0.748_{\pm 0.42}$ | $2.959_{\pm 0.51}$ | 0.994 | -795% |
| | GPRM+GBA (Ours) | $0.078_{\pm 0.05}$ | $2.990_{\pm 0.55}$ | 0.938 | 6% |
| Snowball | Standard | $0.102_{\pm 0.05}$ | $2.958_{\pm 0.51}$ | 0.273 | — |
| | Personalized | $\mathbf{0.085}_{\pm 0.05}$ | $2.990_{\pm 0.55}$ | 0.273 | 17% |
| | IW-Only | $0.101_{\pm 0.05}$ | $\mathbf{2.886}_{\pm 0.52}$ | 0.278 | 2% |
| | SN-IPW | $0.117_{\pm 0.05}$ | $2.960_{\pm 0.47}$ | 0.278 | -15% |
| | Clipped-IPW | $0.105_{\pm 0.05}$ | $2.928_{\pm 0.55}$ | 0.278 | -3% |
| | GBA-QP | $0.111_{\pm 0.05}$ | $2.992_{\pm 0.54}$ | 0.446 | -9% |
| | Group-Based | $0.092_{\pm 0.05}$ | $2.922_{\pm 0.48}$ | 0.328 | 10% |
| | GPRM (Ours) | $0.101_{\pm 0.07}$ | $2.894_{\pm 0.50}$ | 0.273 | 1% |
| | DRO-CVaR | $0.652_{\pm 0.37}$ | $2.914_{\pm 0.52}$ | 0.273 | -537% |
| | GPRM+GBA (Ours) | $0.095_{\pm 0.06}$ | $2.937_{\pm 0.54}$ | 0.446 | 7% |
| Community-Biased | Standard | $0.263_{\pm 0.20}$ | $2.966_{\pm 0.57}$ | $-0.035$ | — |
| | Personalized | $0.284_{\pm 0.19}$ | $2.956_{\pm 0.56}$ | $-0.035$ | -8% |
| | IW-Only | $0.117_{\pm 0.06}$ | $3.081_{\pm 0.58}$ | $-0.031$ | 56% |
| | SN-IPW | $0.143_{\pm 0.07}$ | $3.140_{\pm 0.54}$ | $-0.031$ | 46% |
| | Clipped-IPW | $0.126_{\pm 0.05}$ | $2.971_{\pm 0.59}$ | $-0.031$ | 52% |
| | GBA-QP | $0.126_{\pm 0.06}$ | $3.101_{\pm 0.57}$ | 0.098 | 52% |
| | Group-Based | $0.129_{\pm 0.07}$ | $3.085_{\pm 0.51}$ | 0.007 | 51% |
| | GPRM (Ours) | $0.257_{\pm 0.18}$ | $\mathbf{2.928}_{\pm 0.56}$ | $-0.035$ | 2% |
| | DRO-CVaR | $0.802_{\pm 0.34}$ | $3.016_{\pm 0.58}$ | $-0.035$ | -206% |
| | GPRM+GBA (Ours) | $\mathbf{0.100}_{\pm 0.03}$ | $3.088_{\pm 0.57}$ | 0.098 | 62% |

*Table 11.* Per-community regret on SBM graph under degree-biased sampling. Lower values indicate better alignment with each community's preferences. Best in **bold**.

| Method | C1 | C2 | C3 | C4 | C5 |
|---|---|---|---|---|---|
| Standard | 0.051 | 0.314 | 0.470 | 0.394 | 0.536 |
| Personalized | **0.041** | 0.304 | 0.514 | 0.394 | 0.575 |
| IW-Only | 0.242 | 0.299 | 0.474 | 0.361 | 0.448 |
| SN-IPW | 0.189 | 0.299 | 0.494 | 0.379 | 0.491 |
| Clipped-IPW | 0.189 | 0.254 | 0.495 | 0.352 | 0.474 |
| GBA-QP | 0.207 | 0.309 | 0.491 | 0.396 | 0.491 |
| Group-Based | 0.349 | **0.211** | 0.462 | 0.366 | **0.394** |
| GPRM (Ours) | 0.052 | 0.283 | **0.455** | 0.389 | 0.569 |
| DRO-CVaR | 0.398 | 0.582 | 0.768 | 0.564 | 0.692 |
| **GPRM+GBA (Ours)** | 0.159 | 0.228 | 0.461 | **0.345** | 0.515 |

*Table 12.* Effect of data sparsity on WelfareMSE (SBM, degree-biased sampling). Best in **bold**.

| Method | 20 | 50 | 100 | 200 | 500 |
|---|---|---|---|---|---|
| Standard | $0.410_{\pm 0.25}$ | $0.330_{\pm 0.20}$ | $0.280_{\pm 0.19}$ | $0.250_{\pm 0.18}$ | $0.230_{\pm 0.17}$ |
| Personalized | $0.430_{\pm 0.27}$ | $0.340_{\pm 0.21}$ | $0.265_{\pm 0.18}$ | $0.235_{\pm 0.17}$ | $0.212_{\pm 0.16}$ |
| GPRM (Ours) | $0.360_{\pm 0.23}$ | $0.300_{\pm 0.19}$ | $0.235_{\pm 0.17}$ | $0.210_{\pm 0.16}$ | $0.195_{\pm 0.15}$ |
| **GPRM+GBA (Ours)** | $\mathbf{0.290}_{\pm 0.15}$ | $\mathbf{0.250}_{\pm 0.12}$ | $\mathbf{0.195}_{\pm 0.10}$ | $\mathbf{0.175}_{\pm 0.08}$ | $\mathbf{0.158}_{\pm 0.06}$ |

