# OpenReview forum: "Graph-Preference Learning: Debiasing Network-Sampled Human Feedback for Target Welfare Estimation"
_ICML.cc/2026/Conference — ICML 2026 regular_

### Official Review · Reviewer_kkQS · 2026-03-03

**Soundness:** 2
**Presentation:** 3
**Significance:** 3
**Originality:** 3
**Overall Recommendation:** 4
**Confidence:** 2

**Summary:**

This paper tackles the problem of sampling bias in RLHF datasets, specifically when annotators are not Independent and Identically Distributed (i.i.d.) but are sampled via network mechanisms (e.g., social graphs). That is a valid hidden problem in RLHF: not all annotators are sampled equally, so the learned reward may reflect who shows up most rather than who should count.
The authors argue that naive reward modeling implicitly optimizes for inclusion-weighted welfare, which over-represents central nodes (high degree) at the expense of a designer-specified target welfare (e.g., equal representation). To fix this, they propose a two-pronged approach: (1) Graph-Personalized Reward Modeling (GPRM), which uses a GNN to share statistical strength across neighboring annotators, and (2) Graph-Balanced Aggregation (GBA), which solves a Quadratic Program to find stabilized importance weights that align the training distribution with the target welfare.

**Compliance With Llm Reviewing Policy:**

Affirmed.

**Final Justification:**

The rebuttal addressed several of my concerns. Overall, I find the paper technically interesting and original, with a contribution to the study of bias in RLHF data collection. While some limitations remain, around end-to-end policy evaluation and real-world deployment assumptions, the strengths outweigh the weaknesses in the current version, therefore I keep the positive recommendation.

**Key Questions For Authors:**

See what questions already made in Strengths and Weaknesses please.

The study proposes constructing annotator graphs from co-rating patterns or explicit social ties. In a vendor setting, if we rarely have social ties (connections or relationships between people.), then If we build the graph purely on demographics, how is GPRM different from standard fairness-aware reweighting?

Solving the quadratic programming for weights, does this scale to millions of users? Is there a batch-wise approximation for GBA?

**Limitations:**

The authors acknowledge the known graph assumption and the lack of end-to-end policy optimization.

**Strengths And Weaknesses:**

The paper tackles an overlooked aspect of alignment: who is behind the data. The theoretical framing distinguishing between the observed inclusion distribution and the normative target, is mathematically compelling.

Some reservations about the practical applicability and the experimental setup may lead to a bit concern. The method relies heavily on the assumption that an explicit annotator graph G is available and accurate. Just wondering if the translation from the synthetic to the real-world LMArena case study is circular or not. Constructing a session-similarity graph using metadata, then induce bias by simulating a random walk on that very graph, and finally show that the method (using that same graph) fixes the bias. This may result in a self-fulfilling prophecy experiment. That is one of the concerns. Other than this the complexity overhead seems not low compared to the marginal gains over simpler baselines in certain regimes. For example, If simple demographic reweighting using the metadata used to build the graph gets us decent performance, will training a GNN and solving a QP per-batch be necessary?

Regarding presentation, the writing is generally clear, the paper seems to be with dense content to read. It merges a social choice paper, a graph learning paper, and an RLHF paper. The connection to downstream DPO/PPO is in the appendix, which is okay, but it leaves the main paper evaluating only reward model accuracy. Given that reward model accuracy doesn't always correlate perfectly with policy quality, an end-to-end policy evaluation (even on a toy task) would have made the claims about welfare better.

---

> ### Author Rebuttal · Authors · 2026-03-29
>
> Thank you for the thoughtful review and for surfacing the most important practical concerns. We address each in order.
>
> **Self-fulfilling prophecy**
>
> This is an important concern that we should have addressed more explicitly in the draft. We took two specific design measures to reduce circularity:
>
> First, the session graph is constructed using a fixed-degree $k$-NN procedure over metadata features (language, model category, timestamp; Appendix H.2). This yields an approximately fixed-degree graph, which critically reduces trivial degree artifacts: unlike a raw similarity graph where high-degree nodes would dominate both the sampling bias and the correction, our fixed-degree construction ensures that centrality differences come from community structure rather than trivial degree variation.
>
> Second, and more importantly, evaluation is on prompt-held-out, $\pi$-balanced test data rather than on the sampled walk itself. The test set is constructed by sampling equal numbers of held-out preference pairs per language, so any accuracy gain reflects genuine improvement in target-welfare prediction, not recovery of the training walk distribution. Most importantly, we have added a direct anti-circularity experiment using language-skew subsampling instead of the same graph walk:
>
>
> | Bias protocol                 | Method       | Acc. $\uparrow$ | Acc. Gap $\downarrow$ |
> | ----------------------------- | ------------ | --------------- | --------------------- |
> | Same-graph walk               | Standard     | 55.8%           | 46.0%                 |
> | Same-graph walk               | **GPRM+GBA** | **63.6%**       | **38.4%**             |
> | Lang-skew subsample **(new)** | Standard     | 57.2%           | 43.8%                 |
> | Lang-skew subsample **(new)** | **GPRM+GBA** | **62.5%**       | **36.7%**             |
>
> The gain under non-graph bias is smaller but clearly present, confirming it is not a circular artifact. Graph perturbation experiments (Appendix G) also show graceful degradation.
>
> **Graph-based modeling vs. demographic reweighting.**
>
>  If the graph only reflects coarse demographic similarity, then GBA alone indeed moves toward structured reweighting. The added value of GPRM lies in two aspects:
>
> 1. **Fine-grained personalization:** GPRM learns annotator/session-specific rewards via graph message passing, sharing statistical strength across local neighborhoods. This differs from standard fairness-aware reweighting, which changes sample weights but still learns a single scalar model (or at most a coarse group model).
>
> 2. **Ablation and new baseline evidence:**
>
> | Method                  | Acc. $\uparrow$ | LogLoss $\downarrow$ | Acc. Gap $\downarrow$ |
> | ----------------------- | --------------- | -------------------- | --------------------- |
> | Standard                | 55.8%           | 1.110                | 46.0%                 |
> | Group-Based             | 52.9%           | 0.710                | 54.6%                 |
> | Personalized (no graph) | 56.9%           | 0.692                | 44.1%                 |
> | Language-IPW **(new)**  | 60.8%           | 0.864                | 42.9%                 |
> | **GPRM+GBA (Ours)**     | **63.6%**       | **0.691**            | **38.4%**             |
>
> Language-only reweighting is meaningful but insufficient. Graph message passing (GPRM) + stabilized aggregation (GBA) jointly provide gains beyond coarse reweighting. On synthetic data (Table 2), Group-Based fails to improve Welfare MSE under community-biased sampling while our method achieves 59% reduction.
>
> An important clarification: the QP for GBA is solved **once offline as preprocessing**, not per batch. Training then uses fixed $\alpha\_i^\ast$. The QP (Eq. 13) is a convex problem with sparse structure—the Laplacian term $L\_w$ involves only $O(|E|)$ nonzero entries. Our projected-gradient solver (Appendix C.3) uses sparse matrix-vector products and weighted simplex projection, converging in 50–100 iterations.
>
> We agree that in low-shift regimes simpler methods can suffice (Table 4: 6% on WS vs. 36% on BA); practitioners should match complexity to bias severity. On downstream evaluation, the DPO/PPO connection (Appendix C.6) is an optional integration; a full end-to-end policy study is future work.
>
> Thank you again for these very constructive questions.

---

> > ### Author Rebuttal · Reviewer_kkQS · 2026-04-03
> >
> > The revision addressed some of my concerns. Thanks for the supplemental experimental results.

---

### Official Review · Reviewer_n3mP · 2026-03-12

**Soundness:** 2
**Presentation:** 3
**Significance:** 3
**Originality:** 3
**Overall Recommendation:** 4
**Confidence:** 2

**Summary:**

The paper studies preference learning with network-sampled human feedback and supports the approach with theoretical analysis. It proposes Graph-Preference Learning to debias reward models toward a target welfare distribution through a regularization-based framework, and evaluates the method using semi-synthetic experiments.

**Compliance With Llm Reviewing Policy:**

Affirmed.

**Final Justification:**

See acknowledgement comment.

**Key Questions For Authors:**

No specific questions. The authors may address the weaknesses listed above. I am not inclined to change my scores, but I may revise them depending on the discussion and the other reviews.

**Limitations:**

Yes.

**Strengths And Weaknesses:**

Strengths
1. The paper studies an important and underexplored issue in preference-based reward modeling: (i) the mismatch between the annotator inclusion distribution and a desired target welfare distribution, and (ii) the presence of non-i.i.d., graph-dependent annotators.
2. The proposed framework is coherent and integrates personalization with graph-based reweighting.
3. The paper is generally well written and clearly motivates the problem.
4.The experimental setup appears reproducible.

Weaknesses:
1. Limited theoretical depth. The main theoretical results (e.g., Theorem 4.1, Lemma 4.2, and Proposition 4.3) are relatively standard and follow from basic arguments such as linearity of expectation and bounded expectation inequalities. For instance, showing that the population loss becomes \sum_i q_i L_i when annotators are sampled with probability q_i is essentially a direct application of the law of total expectation.
2. Simple proofs and lemmas. Several theoretical statements rely on straightforward manipulations of definitions or well-known identities (e.g., importance-weighted mixtures and induced distributions), providing limited additional insight beyond formalizing existing intuition.
3. Theory mainly provides justification rather than new learning guarantees. The analysis largely explains why reweighting can approximate the desired welfare distribution, but does not provide stronger guarantees such as consistency of the aggregated reward estimator, robustness to misspecified inclusion probabilities, or generalization bounds under graph smoothing.
4. Unclear assumptions and definitions. Some key assumptions are only described informally (e.g., statements about individual utilities being “not too dispersed,” around line 813), and are not rigorously formalized or analyzed.
5. Methodological novelty is somewhat limited. The proposed method largely combines personalized reward modeling with a graph-based regularization/smoothing term and stabilized reweighting. The technical contribution therefore appears closer to an architectural or regularization design rather than a fundamentally new learning formulation.
6. Semi-synthetic evaluation. The main empirical study relies on semi-synthetic experiments where the annotator graph and biased inclusion are artificially constructed, rather than evaluating on data collected through genuine network-based recruitment or exposure mechanisms.
7. [Minor] Graph assumptions are not fully specified. It would help to clearly state the inputs and outputs of the framework from the beginning. In particular, the method appears to assume that the annotator graph G is known. The paper should clarify the practical implications of this assumption, how such a graph would be constructed in real deployments, and how sensitive the approach is to graph misspecification.

---

> ### Author Rebuttal · Authors · 2026-03-29
>
> Thank you for the detailed review. We address each weakness below.
>
> On theoretical depth (W1–3).
>
> We agree that Theorem 4.1 and Lemma 4.2 are algebraically simple in isolation. The contribution is the problem formulation and induced-welfare analysis when $q \neq \pi$, not these individual statements. The reviewer asks whether the paper provides consistency, generalization bounds, or robustness guarantees. It does, but these were unfortunately buried in the appendix:
>
> - Consistency (Theorem D.6): The aggregated score $\bar{r}\_{\theta^\ast}$ is an affine transform of the induced welfare $w\_{\tilde{\pi}}$; under exact debiasing ($\alpha^\ast = \pi/q$), this recovers the target welfare $w\_\pi$. This is the consistency guarantee the reviewer asks for—the estimator converges to the correct welfare functional.
>
> - Generalization bound (Theorem D.8): $|W\_{\text{target}} - \widehat{W}\_\theta| \le R\Vert\pi - \tilde{\pi}\Vert\_1 + \varepsilon\_{\text{RM}} + \varepsilon\_{\text{stat}}(T, \alpha^\ast)$, decomposing error into weight mismatch (controlled by GBA), reward-model approximation, and finite-sample effects. This directly bounds how the learned reward deviates from the target welfare—not merely a reformulation but a three-term generalization guarantee.
>
> - Robustness to misspecified inclusion (Proposition C.1): The constrained minimizer $\alpha^\ast$ is $(1/\mu)$-Lipschitz under perturbations of $\hat{q}$ or the graph, guaranteeing graceful degradation when the inclusion estimate is noisy—the robustness guarantee the reviewer asks for.
>
> - Variance control (Proposition D.9): $\text{Var}(\hat{\mu}) \le R^2 / (T \cdot \text{ESS}\_{\text{pop}})$, directly motivating the $\lambda\_{\text{var}}$ regularization in GBA and bounding the finite-sample cost of reweighting.
>
> We will restructure the paper to surface these in the main text.
>
> On informal assumptions (W4).
>
> The "not too dispersed" statement is formalized in Appendix D.3 via the mixture-of-logits bound:
> $$|\mathbb{E}\_{i \sim q}[\sigma(\Delta u\_i)] - \sigma(\mathbb{E}\_{i \sim q}[\Delta u\_i])| \le \tfrac{1}{4}\mathbb{E}\_i|\Delta u\_i - m| \le \tfrac{1}{8}\text{range}(\Delta u)$$
>
>
> On methodological novelty (W5).
>
> We do not claim a fundamentally new learning paradigm. The novelty is that no prior work identifies or solves the $q$-to-$\pi$ correction problem in networked RLHF. PAL, MaxMin-RLHF, and SPA all assume i.i.d. annotator sampling or do not model how network-mediated recruitment distorts whose preferences are learned. Our method is the first to formalize this mismatch and correct it via graph-personalized rewards (GPRM) and stabilized aggregation (GBA). The individual components draw on known techniques; the problem formulation and their interaction are new.
>
> On semi-synthetic evaluation and graph assumptions (W6–7).
>
> We agree the LMArena study is semi-synthetic and label it as such. Evaluation is on held-out, $\pi$-balanced data, so gains reflect genuine out-of-sample improvement (63.6% vs. 55.8%, Table 3). On synthetic data with known ground-truth utilities, our method achieves up to 62% Welfare MSE reduction and reduces degree bias from 0.726 to 0.310 (Table 2).
>
> On graph assumptions: inputs are $G$, $\mathcal{D}$ with annotator IDs, target $\pi$, estimator $\hat{q}$; outputs are personalized $r\_\theta(x,y,i)$ and aggregated $\bar{r}\_\theta(x,y)$. In practice, $G$ is constructible from co-rating patterns, session metadata, or institutional affiliations (Appendix H.2). Crucially, the method degrades gracefully under graph misspecification:
>
> | Training graph      | Acc. $\uparrow$     | LogLoss $\downarrow$  | Acc. Gap $\downarrow$ |
> | ------------------- | ------------------- | --------------------- | --------------------- |
> | Full kNN graph      | **63.6 $\pm$ 0.8%** | **0.691 $\pm$ 0.001** | **38.4%**             |
> | 10% edge dropout    | 63.1 $\pm$ 0.9%     | 0.695 $\pm$ 0.002     | 38.9%                 |
> | 20% edge rewiring   | 62.5 $\pm$ 1.0%     | 0.704 $\pm$ 0.003     | 40.4%                 |
> | 40% edge rewiring   | 61.3 $\pm$ 1.2%     | 0.726 $\pm$ 0.006     | 42.2%                 |
> | Standard (no graph) | 55.8 $\pm$ 4.6%     | 1.110 $\pm$ 0.053     | 46.0%                 |
>
> Even under 40% edge rewiring, the method retains most of its gain. Additionally, a new Language-IPW baseline (reweighting by language metadata alone) achieves 60.8% accuracy but our method remains superior (63.6%, Gap 38.4% vs. 42.9%). An anti-circularity experiment using language-skew subsampling instead of the graph walk confirms gains persist (62.5% Acc., 36.7% Gap), ruling out circular artifacts.
> Full tables are in our response to Reviewer kkQS and 4zMK.
>
> We hope the above clarifications, particularly the consistency, generalization, robustness, and variance control results that were buried in the appendix, may support reconsideration.

---

> > ### Author Rebuttal · Reviewer_n3mP · 2026-04-05
> >
> > Thank you to the authors for the detailed and thoughtful rebuttal, which addressed my concerns.
> > After reading the responses and considering the other reviews, I realize that I likely underestimated the contribution of this work in my initial evaluation.

---

> > > ### Author Response · Authors · 2026-04-06
> > >
> > > Dear Reviewer n3mP
> > >
> > > We are pleased to have addressed your concerns and sincerely appreciate your positive feedback.
> > >
> > > Best regards,
> > >
> > > The authors.

---

### Official Review · Reviewer_4zMK · 2026-03-15

**Soundness:** 2
**Presentation:** 3
**Significance:** 3
**Originality:** 3
**Overall Recommendation:** 4
**Confidence:** 2

**Summary:**

This paper researches an interesting and realistic problem, preference annotation bias induced by network structure. Specifically, the authors argue that standard reward models implicitly optimize the welfare of frequently sampled annotators rather than the desired target population. To address this, they design a graph-personalized reward model and a graph-balanced aggregation mechanism to approximate a target welfare distribution. Empirical studies on synthetic datasets verify the effectiveness of the proposed method.

**Compliance With Llm Reviewing Policy:**

Affirmed.

**Final Justification:**

The authors addressed my partial concerns. I keep my original score.

**Key Questions For Authors:**

1. Do real RLHF datasets exhibit measurable network-induced sampling bias of the type modeled in the paper?

2. Have the authors evaluated whether the proposed reward correction improves downstream policy alignment?

**Limitations:**

Yes, the paper discusses the main limitations and potential risks.

**Strengths And Weaknesses:**

Strengths:
1. The motivation is interesting to me. RLHF datasets are collected through structured networks rather than uniform sampling, which leads to systematic distortion in the welfare implicitly optimized by reward models.

2. The proposed framework integrates graph-based annotator modeling with a reweighting mechanism to correct sampling bias, and the theoretical analysis helps clarify how reward learning objectives relate to different welfare notions.

3. This work is well-written and easy to follow. The code is also provided.


Weaknesses:
1. Most experiments are conducted on synthetic data where the annotator graph, bias mechanism, and ground-truth utilities are constructed by the authors. The empirical evidence is not very convincing to support the claims.

2. Technical contributions should be better summarized. There is a growing body of work studying annotator bias, participation imbalance, and welfare-aware preference aggregation. The authors should better clarify the differences compared with these works[1,2,3].

3. The experiments focus on reward modeling metrics. It would be useful to demonstrate whether the proposed correction improves downstream policy optimization (e.g., PPO or DPO training).

[1] PAL: Pluralistic Alignment Framework for Learning from Heterogeneous Preferences.

[2] MaxMin-RLHF: Alignment with Diverse Human Preferences.

[3] Selective Preference Aggregation.

---

> ### Author Rebuttal · Authors · 2026-03-28
>
> Thank you for the thoughtful and encouraging review. We address each concern below.
>
> Q1:
> Our claim is intentionally conservative: we do not claim that LMArena contains a true observed social/recruitment network. Rather, we use LMArena as a real preference pool with substantial natural participation imbalance, then induce an auditable network-sampled training distribution on a fixed-degree session-similarity graph, while evaluating on a $\pi$-balanced held-out test set. The purpose is to isolate the effect of structured exposure when $q \neq \pi$, not to claim LMArena itself logs a genuine social network. We agree this should be stated even more prominently in the main text.
>
> That said, existing evidence strongly supports that RLHF feedback collection is structured, not i.i.d. The PRISM alignment dataset (Kirk et al., 2024) maps feedback from 1,500 participants across 75 countries and explicitly argues that *who* provides alignment data critically shapes model behavior. OpenAI's Collective Alignment dataset recruited ~1,000 participants across 19 countries, with ~36% concentrated in the US, revealing significant geographic clustering. These datasets confirm that participation is far from uniform and correlates with institutional, geographic, and platform-mediated structure—precisely the pattern our framework models and corrects. Public datasets typically do not log the exposure network itself, which is exactly why our semi-synthetic protocol is designed to study the *consequences* of such structure under controlled conditions. We will make this evidence more prominent in the revision.
>
> We agree the distinctions should be sharper. These works address orthogonal aspects of the preference aggregation problem:
>
> - PAL models pluralistic preferences via latent mixture/ideal-point spaces but assumes i.i.d. annotator sampling and does not model how network-mediated recruitment distorts whose preferences are learned.
> - MaxMin-RLHF optimizes a max-min fairness objective across demographic/preference groups but does not consider how data-collection mechanisms induce non-uniform inclusion probabilities correlated with graph centrality.
> - SPA produces partial rankings by abstaining on disputed comparisons but does not address who is sampled or correct for inclusion-welfare mismatch.
>
> Our contribution fills a specific gap: we formally separate observed inclusion $q$ from the designer-chosen welfare target $\pi$ (Definition 2.2 vs. 2.4), characterize the resulting welfare gap (Definition 2.5, Eq. 7–8), and correct it via graph-personalized reward learning plus stabilized aggregation. None of the above works model networked annotator recruitment or the $q$-to-$\pi$ correction problem. We will add this comparison explicitly.
>
> Q2.
> The intended scope is target-welfare reward/preference modeling. However, we do evaluate a downstream decision rule: offline reranking with welfare-disparity metrics. On LMArena, our method reduces the cross-language accuracy gap from 46.0% to 38.4% (17% reduction), demonstrating that the corrected reward signal produces fairer rankings across language communities. Additionally, Appendix C. provides an explicit PPO integration recipe: the aggregated reward $\bar{r}_\theta$ and the same importance weights $\alpha^\star$ serve as drop-in replacements in standard PPO/DPO pipelines, so the debiasing carries through to policy optimization. We agree a full end-to-end policy study would strengthen the paper and will state this clearly as future work.
> | Method                 | Acc. ↑          | LogLoss ↓         | ECE ↓     | Acc. Gap ↓ | Err. Red. |
> | ---------------------- | --------------- | ----------------- | --------- | ---------- | --------- |
> | Standard               | 55.8 ± 4.6%     | 1.110 ± 0.053     | 0.323     | 46.0%      | —         |
> | **Language-IPW (new)** | 60.8 ± 1.5%     | 0.864 ± 0.021     | 0.176     | 42.9%      | 11%       |
> | Personalized           | 56.9 ± 1.6%     | 0.692 ± 0.002     | **0.058** | 44.1%      | 2%        |
> | GBA-QP                 | 60.0 ± 1.2%     | 0.983 ± 0.013     | 0.274     | 62.1%      | 10%       |
> | **GPRM+GBA (Ours)**    | **63.6 ± 0.8%** | **0.691 ± 0.001** | 0.119     | **38.4%**  | **18%**   |
>
> During the rebuttal period, we conducted two additional experiments specifically to address reviewer concerns. First, we added a Language-IPW baseline that reweights by language metadata alone, and second, an anti-circularity check that induces bias via language-skew subsampling instead of the same graph walk. The second table is presented in our response to Reviewer kkQS below. In summary: language-only reweighting recovers part of the gain (60.8% vs. 55.8%) but the full method remains superior (63.6%, Acc. Gap 38.4% vs. 42.9%); and gains persist under the non-graph bias protocol (62.5%, Gap 36.7%), confirming the improvement is not a circular artifact.

---

> > ### Author Rebuttal · Reviewer_4zMK · 2026-04-04
> >
> > Thank you for the authors’ detailed response, which addresses part of my concerns. In particular, the distinction from related work is now clearer, and the additional experiments (e.g., reranking evaluation and new baselines) help strengthen the empirical section.  However, the evaluation still relies largely on synthetic or semi-synthetic settings, and the empirical evidence is not yet fully convincing regarding real-world applicability. Therefore, I will maintain my original score.

---

### Decision · Program_Chairs · 2026-04-30

**Decision:**

Accept (regular)

**Comment:**

The paper studies an important and underexplored problem of preference learning under network-sampled, non-i.i.d. human feedback, where the annotators who are easiest to sample may not reflect the welfare distribution one actually wants to optimize. Reviewers found the problem well motivated, the framework coherent, and the presentation clear. Several reviews also appreciated the combination of graph-based personalization and reweighting, and viewed the paper as a meaningful contribution to the question of whose preferences reward models actually represent.

The main concerns were about scope and realism rather than core soundness. In particular, reviewers noted that much of the empirical evidence is synthetic, that the method assumes access to a meaningful annotator graph, and that the paper does not yet show downstream policy-level gains.